# DeepPCR: Parallelizing Sequential Operations in Neural Networks

**Federico Danieli**   **Miguel Sarabia**   **Xavier Suau**   **Pau Rodríguez**   **Luca Zappella**
Apple
{f_danieli, miguelsdc, xsuaucuadros, pau.rodriguez, lzappella}@apple.com

## Abstract

Parallelization techniques have become ubiquitous for accelerating inference and training of deep neural networks. Despite this, several operations are still performed in a sequential manner. For instance, the forward and backward passes are executed layer-by-layer, and the output of diffusion models is produced by applying a sequence of denoising steps. This sequential approach results in a computational cost proportional to the number of steps involved, presenting a potential bottleneck as the number of steps increases. In this work, we introduce DeepPCR, a novel algorithm which *parallelizes typically sequential operations* in order to speed up inference and training of neural networks. DeepPCR is based on interpreting a sequence of $L$ steps as the solution of a specific system of equations, which we recover using the *Parallel Cyclic Reduction* algorithm. This reduces the complexity of computing the sequential operations from $\mathcal{O}(L)$ to $\mathcal{O}(\log_2 L)$, thus yielding a speedup for large $L$. To verify the theoretical lower complexity of the algorithm, and to identify regimes for speedup, we test the effectiveness of DeepPCR in parallelizing the forward and backward pass in multi-layer perceptrons, and reach speedups of up to $30\times$ for the forward, and $200\times$ for the backward pass. We additionally showcase the flexibility of DeepPCR by parallelizing training of ResNets with as many as 1024 layers, and generation in diffusion models, enabling up to $7\times$ faster training and $11\times$ faster generation, respectively, when compared to the sequential approach.

## 1  Introduction

Neural Networks (NNs) have proven very effective at solving complex tasks, such as classification [26, 14], segmentation [5, 30], and image or text generation [26]. Training NNs, however, is a computationally demanding task, often requiring wall-clock times in the order of days, or even weeks [35, 18], before attaining satisfactory results. Even inference in pre-trained models can be slow, particularly when complex architectures are involved [4]. To reduce training times, a great effort has been invested into speeding up inference, whether by developing dedicated software and hardware [7, 22, 23], or by investigating algorithmic techniques such as (early) pruning [28, 40, 20, 27, 43, 9].

Another possibility for reducing wall-clock time, and the one we focus on in this work, consists in parallelizing computations that would otherwise be performed sequentially. The most intuitive approach to parallelization involves identifying sets of operations which are (almost entirely) independent, and executing them concurrently. Two paradigms that follow this principle are *data-parallelization*, where multiple datapoints are processed simultaneously in batches; and *model-parallelization*, where the model is split among multiple computational units, which perform their evaluations in parallel [1].

Still, certain operations which are key for training and inference in NNs have a sequential structure. The forward and backward pass of a NN are examples of such operations, where activations

37th Conference on Neural Information Processing Systems (NeurIPS 2023).

(or gradients) are computed sequentially, one layer at a time. Moreover, some generative models suffer from similar shortcomings: in diffusion models (DMs), for example, the output image is generated through a sequence of denoising steps [36]. Sequential operations such as these require a computational effort which grows linearly with the sequence length $L$ (that is, with the number of layers, or denoising steps), which represents a bottleneck when $L$ is large. Given the prevalence of these operations, any effort towards their acceleration can result in noticeable speed gains, by drastically reducing training and inference time. Further, faster computations may allow exploration of configurations which were previously unfeasible due to the excessive time required to perform these operations sequentially: for example, extremely deep NNs, or diffusion over tens of thousands of denoising steps.

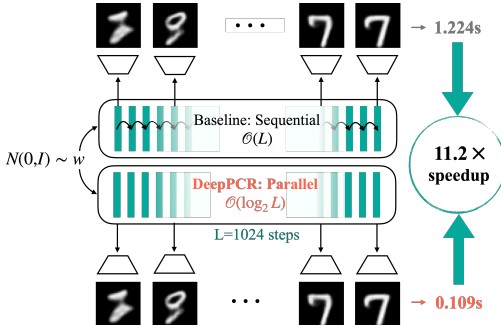

Figure 1: DeepPCR allows executing sequential operations, such as denoising in latent diffusion, in $\mathcal{O}(\log_2 L)$ time, as opposed to the $\mathcal{O}(L)$ needed for the traditional approach ($L$ being the number of steps). In our experiments, DeepPCR achieves a $11.2\times$ **speedup for image generation with latent diffusion** with respect to the sequential baseline, with comparable quality in the recovered result.

In this work we introduce DeepPCR, a novel method which provides a flexible framework for turning such sequential operations into parallel ones, thus accelerating operations such as training, inference, and the denoising procedure in DMs.

The core idea behind DeepPCR lies in interpreting a sequential operation of $L$ steps as the solution of a system of $L$ equations, as illustrated in Sec. 2. DeepPCR assumes the output of each step only depends on that of the previous one, that is, the sequence satisfies the Markov property. If this holds, we can leverage the specific structure of the resulting system to tackle its solution in parallel, using the Parallel Cyclic Reduction algorithm (PCR) [10, 2]. This algorithm, described in Sec. 3, guarantees the recovery of the solution in $\mathcal{O}(\log_2 L)$ steps, rather than the $\mathcal{O}(L)$ steps required for its sequential counterpart. In our test, this translates into inference speedups of up to $30\times$ for the forward pass and $200\times$ for the backward pass in certain regimes, and $11.2\times$ speedup in image generation via diffusion, as shown in Fig. 1. The reduced computational complexity comes in exchange for higher memory and computational intensity. Therefore, in Sec. 4.1 we investigate in detail regimes for speedup, as well as the trade-off between our method and the sequential approach, considering as model problems the forward and backward passes through multi-layer perceptrons (MLPs) of various sizes. In Sec. 4.2 we then observe how this translates into speedups when training ResNet architectures. Finally, in Sec. 4.3 we showcase how DeepPCR can be applied to accelerate other types of sequential operations as well, choosing as example the denoising procedure in DMs.

**Previous Work**   The idea of parallelizing forward and backward passes through a DNN was spearheaded in [13, 32, 24, 31, 41], under the concept of *layer-parallelization*. For the most part, these approaches have been limited to accelerating the training of deep ResNets [15], since they rely on the interpretation of a ResNet as the discretization of a time-evolving differential equation [6], whose solution is then recovered in a time-parallel fashion [11].

More closely resembling our approach is the work in [39], where the authors start by interpreting a sequential operation as the solution of a large system of equations, which is then targeted using parallel solvers. They too focus on accelerating forward and backward passes on ResNets, but also consider some autoregressive generative models (specifically, MADE [12] and PixelCNN++ [38]), similarly to what is done in [44]. The main difference between our approach and the one in [39] lies in the solvers used for tackling the target system in parallel. They rely on variations of Jacobi iterations [34], which are very cost-efficient, but "fall short when the computational graph [of the sequential operation considered] is closer to a Markov chain" [39]: we can expect the convergence of Jacobi to fall to $\mathcal{O}(L)$ in that case, thus providing no speedup over the sequential approach. By contrast, our method specifically targets Markov sequences, solving them with complexity $\mathcal{O}(\log_2 L)$, and is in this sense complementary to theirs. We point out that a similar theoretical foundation for our method was proposed in [33], however it was not verified experimentally, nor has it been considered for applications other than forward and backward passes acceleration.

**Main Contributions**  The main contributions of this work can be summarized as follows:

i) We propose DeepPCR, a novel algorithm for parallelizing sequential operations in NN training and inference, reducing the complexity of these processes from $\mathcal{O}(L)$ to $\mathcal{O}(\log_2 L)$, $L$ being the sequence length.

ii) We analyze DeepPCR speedup of forward and backward passes in MLPs, to identify high-performance regimes of the method in terms of simple architecture parameters, and we discuss the trade-offs between memory consumption, accuracy of the final solution, and speedup.

iii) We showcase the flexibility of DeepPCR applying it to accelerate training of deep ResNet [15] on MNIST [8], and generation in Diffusion Models trained on MNIST, CIFAR-10 [25] and CelebA [29]. Results obtained with DeepPCR are comparable to the ones obtained sequentially, but are recovered up to $7\times$ and $11\times$ faster, respectively.

## 2  Turning sequential operations into systems of equations

Our approach is rooted in casting the application of a sequence of $L$ steps as the solution of a system of $L$ equations, which we then proceed to solve all at once, in parallel. In this section, we illustrate a general framework to perform this casting and recover the target system. Specific examples for the applications considered in our work (namely forward and backward passes, and generation in diffusion models) are described in appendix A. The algorithm for the parallel solution of the recovered system is outlined in Sec. 3.

Consider a generic sequence of steps in the form $z_l = f_l(z_{l-1})$, for $l = 1, \ldots, L$, starting from $z_0 = f_0(x)$. The various $f_l$ could represent, for example, the application of layer $l$ to the activations $z_{l-1}$ (if we are considering a forward pass), or the application of the $l$-th denoising step to the partially recovered image $z_{l-1}$ (if we are considering a diffusion mechanism). Notice we are assuming that the output of each step $z_l$ depends only on that of the previous step $z_{l-1}$ and no past ones: that is, we are considering sequences that satisfy the *Markov* property (a discussion on the limitations related to this assumption, and possible workarounds to relax it, is provided in appendix B). We can collate this sequence of operations into a system of equations for the collated variable $z = [z_0^T, \ldots, z_L^T,]^T$, and obtain:

$$
\mathcal{F}(z) = \begin{bmatrix} z_0 - f_0(x) \\ z_1 - f_1(z_0) \\ \vdots \\ z_L - f_L(z_{L-1}) \end{bmatrix} = \begin{bmatrix} I & & & \\ -f_1(\cdot) & I & & \\ & \ddots & \ddots & \\ & & -f_L(\cdot) & I \end{bmatrix} \begin{bmatrix} z_0 \\ z_1 \\ \vdots \\ z_L \end{bmatrix} - \begin{bmatrix} f_0(x) \\ 0 \\ \vdots \\ 0 \end{bmatrix} = 0.
$$

(1)

Notice that, to better highlight the structure of the operator involved, we are abusing matrix notation and considering that the "multiplication" of $f_l(\cdot)$ with $z_{l-1}$ results in its application $f_l(z_{l-1})$, although $f_l$ is generally a nonlinear operator. To tackle the nonlinearity (when present), we use Newton's method [34]. In more detail, denoting with a superscript $k$ the Newton iteration, we start from an initial guess for iteration $k = 0$, namely $z = z^0$, and iteratively update the solution $z^{k+1} = z^k + \delta z^k$ by solving the linearized system

$$
J_\mathcal{F}|_{z^k} \, \delta z^k = -\mathcal{F}(z^k),
$$

(2)

until we reach convergence. $J_\mathcal{F}|_{z^k}$ denotes the Jacobian of the global sequential operation $\mathcal{F}(z)$ evaluated at the current iteration $z^k$. This Jacobian defines the target system we need to solve, and obeys a very specific structure: taking the derivative of (1) with respect to $z$, and expanding (2), we see that

$$
(2) \iff \begin{bmatrix} I & & & \\ -J_{f_1}|_{z_0^k} & I & & \\ & \ddots & \ddots & \\ & & -J_{f_L}|_{z_{L-1}^k} & I \end{bmatrix} \begin{bmatrix} \delta z_0^k \\ \delta z_1^k \\ \vdots \\ \delta z_L^k \end{bmatrix} = \begin{bmatrix} f_0(x) - z_0^k \\ f_1(z_0^k) - z_1^k \\ \vdots \\ f_L(z_{L-1}^k) - z_L^k \end{bmatrix},
$$

(3)

that is, the system is *block bidiagonal*. This structure is a direct consequence of the Markovian nature of the sequential operation: since each step relates only two adjacent variables $z_{l-1}$ and $z_l$, only two diagonals appear. The core of DeepPCR lies in applying a specialized parallel algorithm for solving systems with this very structure, as described in Sec. 3.

# 3 Parallel Cyclic Reduction for NNs

The solution of a block bidiagonal system is usually obtained via forward substitution: once $z_l$ is known, it is used to recover $z_{l+1}$ and so on, in increasing order in $l$. This procedures is efficient, but inherently sequential, and as such might represent a bottleneck for large $L$. Interestingly, there exist alternative algorithms for the solution of such systems, which trade-off more complex instructions and extra memory consumption for a higher degree of parallelization. One such algorithm, and the one our method is based on, is Parallel Cyclic Reduction (PCR) [19]. Originally, PCR was devised to parallelize the solution of tridiagonal systems; in this work, we describe its adaptation for bidiagonal systems such as (3). In a nutshell, PCR works by combining the equations of a system to progressively reduce its dimension, until it becomes easily solvable. Pseudo-code for the adapted algorithm is reported in Alg. 1, and a schematic of how the reduction is performed is outlined in Fig. 2. More details on its functioning are provided next.

We start by noting that systems like (3) can be compactly represented as a set of equations involving only two *adjacent* variables $\delta z_{l-1}, \delta z_l$:

$$\delta z_l - \underbrace{J_{f_l}|_{z_{l-1}}}_{=:A_l^0} \delta z_{l-1} - \underbrace{(f_l(z_{l-1}) - z_l)}_{=:r_l^0} = 0, \qquad l = 1, \ldots, L, \tag{4}$$

with $\delta z_0 = f_0(x) - z_0^k$ known. The 0 superscripts in the operators $A_l^0$ and vectors $r_l^0$ defined above refer to the current (0-th) PCR step. As a first step for PCR, we substitute the $(l-1)$-th equation into the $l$-th, for each $l$ in parallel, recovering

$$\delta z_l - \underbrace{A_l^0 A_{l-1}^0}_{=:A_l^1} \delta z_{l-2} - \underbrace{\left(r_l^0 - A_l^0 r_{l-1}^0\right)}_{=:r_l^1} = 0, \qquad l = 2, \ldots, L. \tag{5}$$

Notice that the original structure is still preserved, but now the equations relate variables $l$ to $l-2$. In other words, the even and the odd variables have become separated, and we have split the original system into two independent subsystems: one involving variables $\delta z_0, \delta z_2, \ldots$, the other $\delta z_1, \delta z_3, \ldots$ At the next step, we substitute equations $l-2$ into $l$, to recover:

$$\delta z_l - \underbrace{A_l^1 A_{l-2}^1}_{=:A_l^2} \delta z_{l-4} - \underbrace{\left(r_l^1 - A_l^1 r_{l-2}^1\right)}_{=:r_l^2} = 0, \qquad l = 5, \ldots, L, \tag{6}$$

so that now only variables at distance 4 are related. Ultimately, at each step of PCR, we are splitting each subsystem into two independent subsystems. If we iterate this procedure for $\log_2 L$ steps, we finally obtain $L$ systems in one variable, which are trivially solvable, thus recovering the solution to the original system.

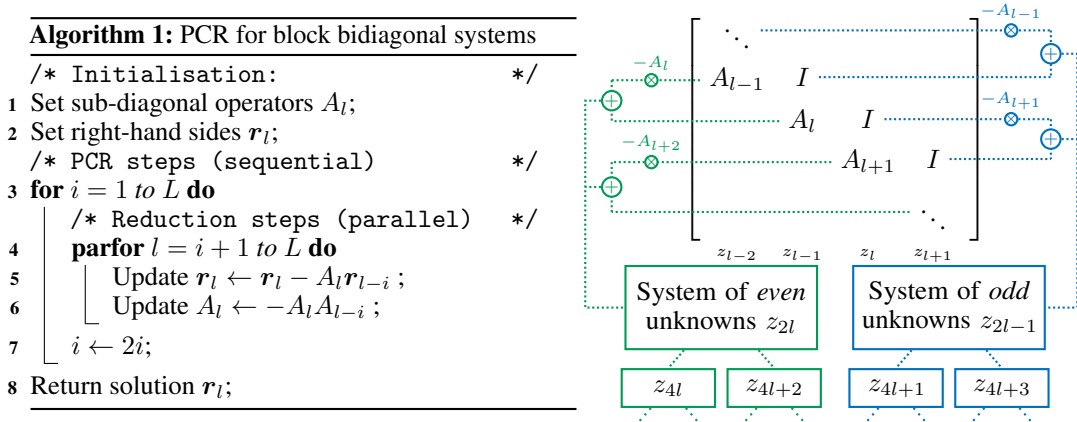

Figure 2: Left: pseudo-code for PCR algorithm. Right: schematic of row reductions in PCR: green rows are combined pairwise to obtain a system of equations in even unknowns; at the same time, blue rows are combined to obtain a system in odd unknowns only. The result is two independent systems with half the original number of unknowns. The procedure is then repeated for $\log_2 L$ steps.

### 3.1 Limitations of DeepPCR

The main advantage of using DeepPCR for solving (1) lies in the fact that it requires only $\mathcal{O}(\log_2 L)$ sequential steps, as opposed to the $\mathcal{O}(L)$ necessary for traditional forward substitution. However, some conditions must be verified for this procedure to be effective in achieving speedups. We discuss next some recommendations and limitations associated with DeepPCR.

**Effective speedup for deep models** While PCR requires fewer sequential steps overall, each step is in principle more computationally intensive than its sequential counterpart, as it requires multiple matrix-matrix multiplications to be conducted concurrently (by comparison, one step of the sequential case requires applying the step function $f_l(z)$), as per line 6 in Alg. 1. If this cannot be done efficiently, for example because of hardware limitations, then we can expect performance degradation. Moreover, the difference between the linear and logarithmic regimes becomes useful only for large $L$. Both these facts are investigated in Sec. 4.1.

**Controlling Newton iterations** Whenever (1) is nonlinear, the complexity actually becomes $\mathcal{O}(c_N \log_2 L)$, where $c_N$ identifies the number of Newton iterations necessary for convergence. On the one hand, it is important for $c_N$ to remain (roughly) constant and small, particularly with respect to $L$, for the logarithmic regime to be preserved and speedups to be attained; on the other hand, there is a positive correlation between $c_N$ and the accuracy of the solution recovered by the Newton solver. Implications of this trade-off are discussed in Sec. 4.4. We also point out that, in general, Newton's method provides no guarantees on *global* convergence (unlike Jacobi's in [39], which reduces to the sequential solution in the worst-case scenario). Even though in our experiments the method never fails to converge, it is worth keeping in mind that ultimately the solver performance is dependent both on the regularity of the target function (1), and on the initialization choice. In particular, the effect of the latter is investigated in appendix F, but already the simple heuristics employed in our experiments (such as using the average of the train set images as initialization for the output of our DMs) have proven to be effective in providing valid initial guesses for Newton.

**Benefits from larger memory** To apply DeepPCR, it is necessary to store the temporary results from the equation reductions (most noticeably, the operators $A_l$ in line 6 in Alg. 1). The associated memory requirements scale linearly in the number of steps $L$ and quadratically in the dimension of each step output $z$. This results in an increase in memory usage with respect to classical approaches (roughly $2\times$ as much for forward passes in MLPs, as measured and reported in appendix C.2). We point out that the additional memory requirements of DeepPCR may limit its applications to some distributed training settings where memory is already a bottleneck. Moreover, one can expect additional communication overhead to arise in these settings.

## 4 Results

In this section, we set out to demonstrate the applicability of DeepPCR to a variety of scenarios. We start by investigating the performance characteristics of DeepPCR when applied to the forward and backward passes through a Multi-Layer Perceptron (MLP). Experimenting with this model problem is mostly aimed at identifying regimes where DeepPCR achieves speedup. Specifically, in Sec. 4.1 we show that, when applied to the forward pass, DeepPCR becomes effective in architectures with more than $2^7$ layers. For the backward pass, this regime is reached earlier, in architectures with $2^5$ layers. Next, we explore the effects of applying DeepPCR to speedup the whole training procedure, considering ResNets architectures: in Sec. 4.2 we verify not only that the speedups measured for the single forward and backward passes carry over to this scenario, achieving a $7\times$ speedup over the sequential implementation, but also that training with DeepPCR results in equivalent models than using sequential passes. In Sec. 4.3, we showcase the flexibility of DeepPCR by using it to speedup another type of sequential operation: the denoising procedure employed by diffusion models in image generation. We consider applications to latent diffusion, and find speedups of up to $11.2\times$, with negligible error with respect to the sequential counterpart. Lastly, in Sec. 4.4 we focus on the role of the Newton solver in the DeepPCR procedure, establishing that the method remains stable and recovers satisfactory results even by limiting the number of Newton iterations, thus allowing to trade-off additional speedup for an increased approximation error with respect to sequential solutions.

All the experiments in this section were conducted on a V100 GPU with 40GB of RAM; our models are built using the PyTorch framework, without any form of neural network compilation.

## 4.1 Speeding up forward and backward passes in MLPs: identifying performance regimes

Our first goal is to identify under which regimes DeepPCR can effectively provide a speedup. To this end, we consider a single forward pass through a randomly initialized MLP with a constant number of hidden units (namely, its width $w$) at each layer, and profile our algorithm for varying $w$ and NN depth, $L$. Notice that these two parameters directly affect the size of (3): $L$ determines the number of equations, while $w$ the unknowns in each equation; as such, they can be used as indication of when to expect speedups for more complex problems.

Timing results for these experiments are reported in Fig. 3. The leftmost column refers to the sequential implementation of forward (top) and backward (bottom) pass, and clearly shows the linear complexity in $L$ of such operations: the curves flatten on a line of inclination 1. Conversely, the graphs in the middle column illustrate DeepPCR's performance, and trace a logarithmic curve for the most part, confirming the theoretical expectations on its $\mathcal{O}(\log_2 L)$ complexity. Notice this reduces the wall-clock time for a single forward pass from $0.55s$ to $0.015s$, and for a backward pass from $589ms$ to $2.45ms$, corresponding to speedups of $> 30\times$ and $200\times$, respectively, at least for the most favorable architectures - and this despite the fact that there has been more than 20 years of optimization into extracting the best performance from the current GPU hardware when running the sequential forward and backward pass. This result is encouraging as our proposed algorithm can gain from further optimization in each of its steps.

As the MLP grows in width, however, the logarithmic regime is abandoned in favour of a linear regime. This performance degradation is due to the fact that the reductions in line 6 necessary for PCR cannot be performed concurrently anymore. Notice that $w$ relates directly to the size of the Jacobian blocks in (3), so we can expect similar problems whenever the Jacobian size grows past a given threshold. This issue is caused by hardware limitations, and can be addressed by using dedicated hardware or by optimizing the implementation: evidence of this claim is provided in appendix C.1, where we measure how the threshold for abandoning the logarithmic regime shifts as we use GPUs with different amounts of dedicated memory. Finally, the rightmost graphs in Fig. 3 show the ratio of timings for the sequential versus parallel implementation: any datapoint above 1 indicates effective speedup. The break-even point between the two methods lies around $L \approx 2^7$ for the forward pass.

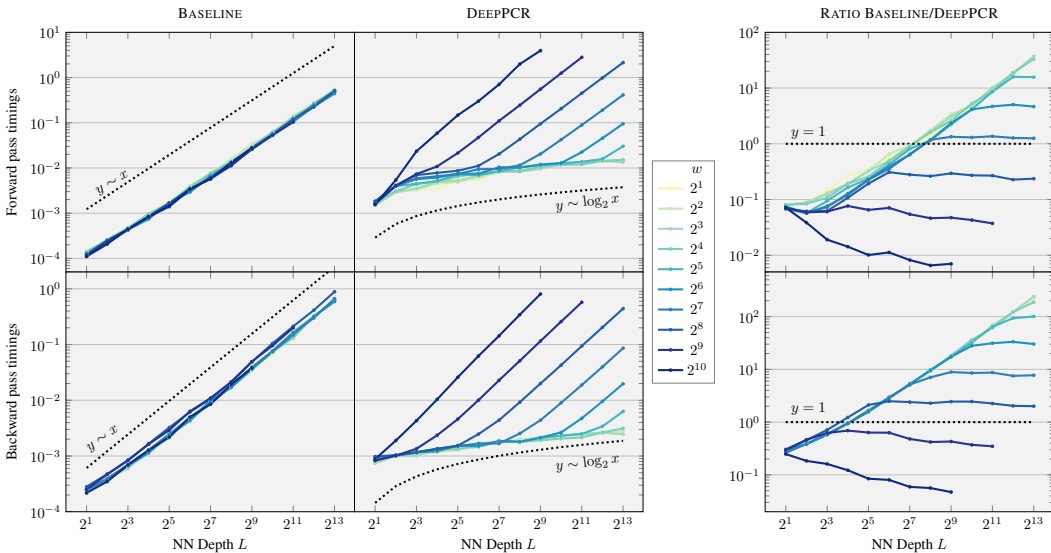

Figure 3: Time to complete a single forward pass (top) and backward pass (bottom), for MLPs of varying depths $L$ and widths $w$, with ReLU activation function. Each datapoint reports the minimum time over 100 runs. The left, center, and right columns refer to the sequential implementation, the DeepPCR implementation, and the ratio between the timings of the two, respectively.

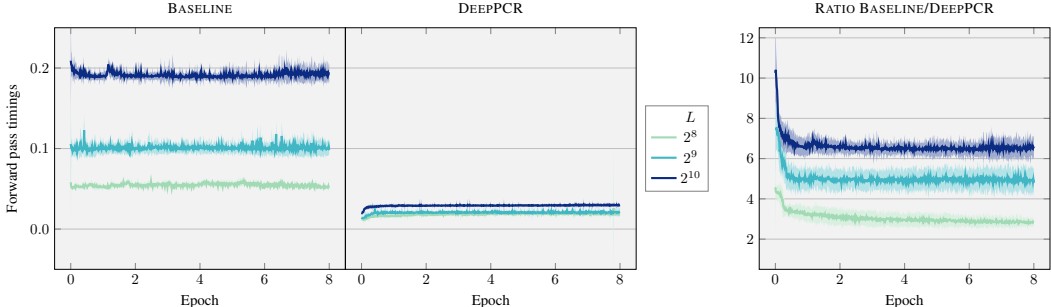

Figure 4: Time to complete forward pass during training, for sequential (left) and DeepPCR implementation (center), and ratio between the two (right), for ResNets of varying depths $L$, with $w = 2^4$, skip connection of length 4, and ReLU activation function. Each datapoint is an average over 100 optimization steps, and the shaded area spans to $\pm 1$ standard deviation.

Results for backward pass are qualitatively comparable, but achieve break-even at $L \approx 2^5$: this gain is due to the fact that the backward pass is a linear operation, and as such does not require Newton iterations. For a more in-depth analysis of the role of the Newton solver, we refer to Sec. 4.4.

### 4.2 Speeding up training of ResNets

The results in Sec. 4.1 identify regimes where one can expect to achieve speedup using DeepPCR, but they only refer to a single forward and backward pass through a freshly initialized model. The results in this section aim to verify that DeepPCR can be used to accelerate forward and backward passes for the whole training procedure, and that the speedup is maintained throughout. To this end, we train a deep ResNet model composed of only fully-connected layers. Each ResNet block consists of 4 layers of width $2^4$ and the ReLU activation function. The models are trained on a classification task on MNIST [8], both using the sequential approach and DeepPCR. We train for 8 epochs using an SGD optimizer with a learning rate of $10^{-3}$ without a scheduler. We perform training runs with various seeds but report results from only one for readability: the others are comparable, and we show their statistics in appendix D. In Fig. 4 we report the evolution of the wall-clock time measurements for the forward pass throughout the training procedure. We can notice these remain roughly constant, confirming that the speedup achieved by DeepPCR is preserved during training. Notice that using DeepPCR translates into a speedup of $7\times$ over the sequential implementation: over the whole course of training, this entails a wall-clock time difference of $3.2h$ versus $30min$, even without including the gains from the backward pass.

As mentioned in Sec. 3.1, we remind the reader that DeepPCR uses Newton in order to solve (1). Being Newton an approximate solver, one may wonder whether we are accumulating numerical errors with respect to the sequential solution, how does it affect the evolution of the parameters, and what is

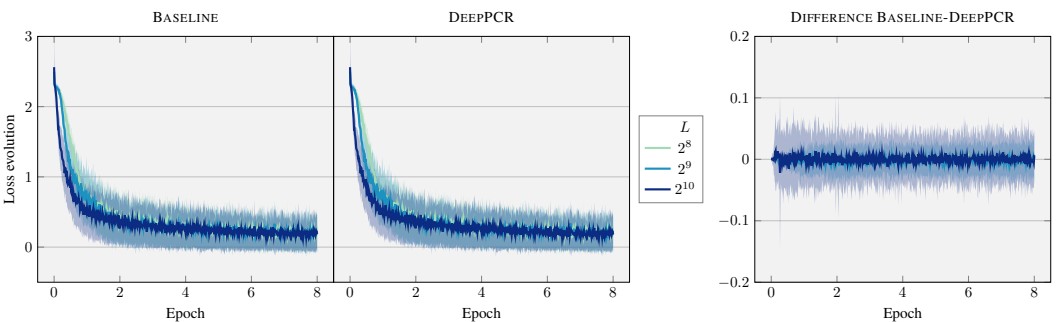

Figure 5: Loss evolution during training with forward and backward passes computed sequentially (left), with DeepPCR (center), and difference between the two (right), for ResNets of varying depths $L$, with $w = 2^4$, skip connection of length 4, and ReLU activation function. Each datapoint is an average over 100 optimization steps, and the shaded area spans $\pm 1$ standard deviation.

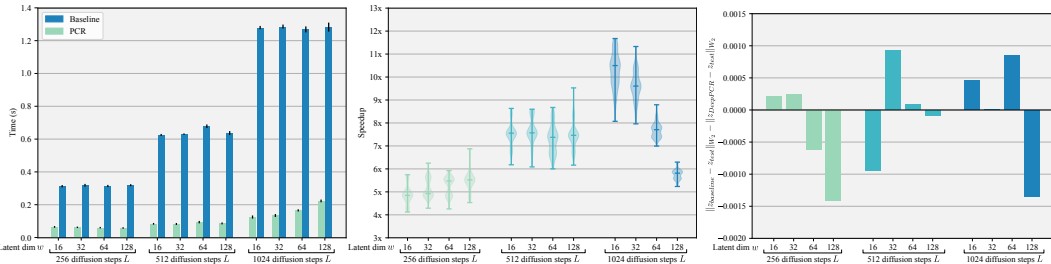

Figure 6: Results from applying DeepPCR to speedup image generation in latent diffusion trained on MNIST, for various latent space dimensions $w$ and number of denoising steps $L$. Left: timings using sequential and DeepPCR approaches (average over 100 runs). Middle: violin plots of speedups distribution (ratio of sequential/DeepPCR timings for 100 runs). Right: difference between Wasserstein-2 distances to test distribution of latents recovered sequentially and using DeepPCR.

the impact on the quality of the final trained model. In our experiments, we measure such impact by comparing the evolution of the loss curves for the models trained sequentially and in parallel with DeepPCR. These are reported in Fig. 5, which shows that, for our experiments, the evolutions are practically equivalent. To further confirm this, we report the accuracy evolution on the test set in appendix D: in both cases, it sits around $94\%$ at the end of training. The effects of the Newton solver on performance are further discussed in Sec. 4.4.

## 4.3 Speeding up image generation in Diffusion Models

The experiments in this section showcase the flexibility of DeepPCR in accelerating more general definitions of sequential operations. As an example, we apply DeepPCR to speedup image generation via latent-space diffusion models [37]. Note that we are interested in parallelizing the whole denoising procedure, rather than the single forward pass through the denoiser: we refer to appendix A.4 for the specifics on how this operation falls within the DeepPCR framework. We consider the size of the latent space and the number of denoising steps as the two main parameters which can impact the effectiveness of DeepPCR, and measure how the performance of our method varies according to them. Notice that, in determining the size of system (3), these two parameters cover the same role as $w$ and $L$ in Sec. 4.1, respectively, so we identify them using the same notation. Our latent diffusion model considers a simplification of the KL-AutoEncoder introduced by [37] as an encoder, and a custom MLP with residual connections as denoiser: see appendix E for details.

In Fig. 6 (left) we report the average time[1] for completing the diffusion procedure, either sequentially or using DeepPCR, for 100 runs on architectures trained on MNIST with various values of $w$ and $L$. Notice how even in this case the time for the sequential approach grows linearly with respect to the number of denoising steps, while for DeepPCR the growth is logarithmic for the most part. Increasing $w$ past $\sim 2^6$, though, results in a speedup reduction for the largest $L = 2^{10}$, matching what is observed in Fig. 3: similarly, this is related to hardware limitations, and we refer again to appendix C.1 for an analysis of the phenomenon. The distributions of the associated speedups are also plotted in Fig. 6 (middle), where we can see that DeepPCR manages to generate images up to $11\times$ faster, reducing the required time from $1.3s$ to $0.12s$ for certain configurations. To ensure the quality of the resulting images, we follow the FID score [16] and measure the Wasserstein-2 distance between the latent distribution of the original test set and the latent distribution of the images recovered, either sequentially or using DeepPCR. The difference of these distances is also reported in Fig. 6, and is consistently close to 0, hinting that using either method results in images of similar qualities. Some examples images generated sequentially or using DeepPCR can be seen in Fig. 18, to further confirm that they are hardly distinguishable. We also experimented with diffusion in pixel-space: the corresponding timings can be found in Tab. 2, and their behavior mimics what was observed for latent diffusion.

---

[1]We point out that the timings in Fig. 6 and 7 are a proxy, evaluated assuming perfect parallelizability of the Jacobian assembly operation necessary to initialize system (3). We could not measure exact wall-clock time due to incompatibilities between the `vmap` and `autograd` functionalities provided in PyTorch. Nonetheless, this proxy is reasonably accurate, as the time required to assemble the Jacobians is negligible with respect to that for the PCR reduction (see appendix E.2, and particularly Fig. 17 for details).

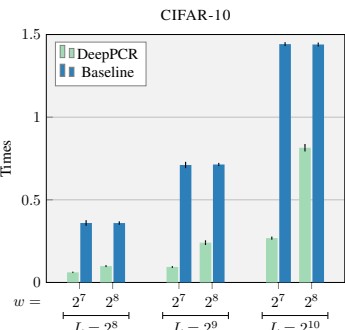
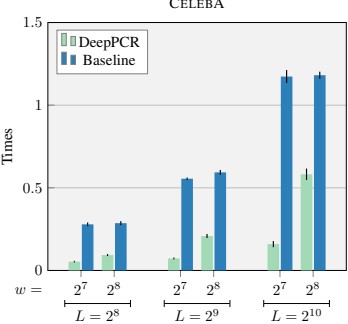

Figure 7: Results from applying DeepPCR to speedup image generation in latent diffusion, for various latent space dimensions $w$ and number of denoising steps $L$. The timings compare sequential (baseline) and DeepPCR approaches, reporting an average over 100 runs, for models trained over the CIFAR-10 (left) and CelebA (right) datasets.

Finally, in order to provide empirical evidence of the capability of DeepPCR to provide speedup also for other datasets, we experiment with latent diffusion on CIFAR-10 [25] and CelebA [29] as well. The corresponding timings results are reported in Fig. 7. We limit ourselves to $w > 2^6$ due to the difficulty of training VAEs for these datasets on smaller latent dimensions. Nonetheless, the timing results are comparable to the ones measured for MNIST in Fig. 6, and even in this case we manage to recover speedups of $8\times$ and $9\times$ for CIFAR-10 and CelebA, respectively. We can see that also for these more complex datasets the performance of DeepPCR starts degrading for $w > 2^7$, similarly to what is observed in Fig. 6. This observation further confirms that the speedup attained by DeepPCR is influenced by the problem parameters $w$ and $L$, but is otherwise dataset-independent.

## 4.4 Accuracy/Speedup trade-off: analysis on Newton convergence

As outlined in Sec. 2, when system (1) is nonlinear, DeepPCR relies on a Newton solver. This is an iterative solver, which only recovers an *approximate* solution, correct up to a fixed tolerance. The experiments in the previous sections were conducted with a tolerance of $10^{-4}$, as we were interested in recovering a solution which would closely match the sequential one. The tolerance of the solver, however, grants us a degree of freedom in trading off accuracy for additional speedup. In this section we investigate in detail the properties of the Newton method when used for the solution of the problems considered in Sec. 4.1 and 4.2.

As a first result, we show that Newton can indeed recover high-quality solutions, within a number of iterations $c_N$ which is small and roughly independent of the configuration considered. To this purpose, we report in Fig. 8 the values of $c_N$ recorded for the experiments in Sec. 4.1 and 4.2. In all configurations considered, they remained bounded below $c_N \leq 6$, and practically independent on the system configuration, particularly of $L$. In Fig. 8 (first on the left), we see that the performance of the Newton solver is indeed impacted by the type of activation function used in the layers of the MLP: using ReLUs generally requires more iterations for convergence than using a smoother counterpart such as sigmoid. This is in line with the properties of the Newton method which assumes differentiability of the underlying function for fast convergence.

Additionally, for the same set-up, we show (second plot in Fig. 8) the error between the solution recovered via Newton with DeepPCR and the traditional solution, recovered sequentially. This error is expressed in terms of the $L^2$ difference of the NN output (for the experiments in Sec. 4.1) and in terms of the $L^\infty$ difference of the parameters evolution (for the experiments in Sec. 4.2), to better reflect the relevant metrics of the two experiments. The former sits almost always around machine precision, confirming that sequential and DeepPCR solutions are extremely close. For the latter, we see that small numerical errors eventually accumulate throughout the training procedure. Still, the discrepancies are bounded, and this does not affect the final performance of the trained model (as shown also in Fig. 5, and appendix D).

Finally, we conduct an ablation study on the effect of reducing the accuracy of the recovered solution. To this end, we consider again the framework in Sec. 4.2, but this time we fix the number of Newton iterations for solving the forward pass to increasingly small values, and check at which stage training

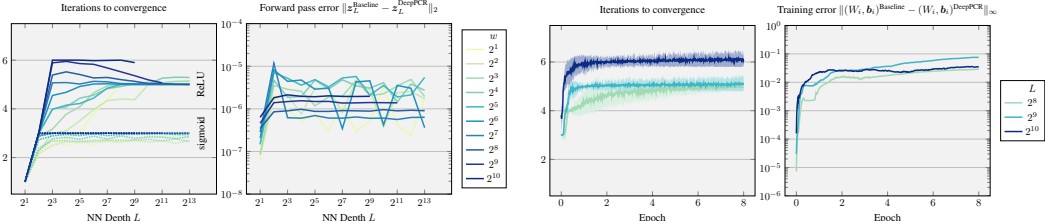

(a) Newton solver analysis for forward pass through an MLP of depth $L$ with layers of width $w$ (same configurations as in Fig. 3). Left plot indicates the number of iterations until convergence. Right plot indicates the error between the model output obtained via sequential and DeepPCR approaches.

(b) Newton solver analysis for ResNet training with $L$ layers (same configuration as in Fig. 5). Left plot indicates number of iterations until convergence (mean across 100 optimization steps), and shaded area spans $\pm 1$ standard deviation. Right plot indicates evolution of error between network parameters obtained training with sequential baseline and DeepPCR.

Figure 8: Newton solver analysis for forward pass through MLP (left), and ResNet training (right).

of the ResNets fails. The results reported in appendix F.1 show that, for the problem considered, stopping Newton at $c_N = 3$ still results in successful training. This translates into an additional $2\times$ speedup with respect to the ResNet times reported in Fig. 4, for a total of up to $14\times$ speedup. For more general problems, we can expect that fine-tuning the Newton solver would play a relevant role in the final speedup attained. Particularly, choosing the correct initial guess for the system and identifying the most apt tolerance level.

## 5 Conclusion, Limitations, and Future Work

We introduced DeepPCR, a method for parallelizing sequential operations which are relevant in NN training and inference. The method relies on the target sequence being Markovian: if this is satisfied, the sequential operation can be interpreted as the solution of a bidiagonal system of equations. The system is then tackled using Parallel Cyclic Reduction, combined with Newton's method. We investigated the effectiveness and flexibility of DeepPCR by applying it to accelerate: i) forward/backward passes in MLPs, ii) training of ResNets, and iii) image generation in diffusion models, attaining speedups of up to $30\times$, $7\times$, and $11\times$ for the three problems, respectively. We identified regimes where the method is effective, and further analyzed trade-offs in terms of speedup, accuracy, and memory consumption.

The main bottleneck for our DeepPCR implementation is represented by the decay in performance associated with the growth in size of the Jacobian blocks in (3). While this can be curbed by using hardware with larger memory and/or better parallelization capabilities, investigating alternative ways to circumvent this issue would greatly benefit the applicability of DeepPCR. Another potential issue is related to the reliance of DeepPCR on a Newton solver for recovering the solution to the target system. While Newton proved to be reasonably robust for the target applications we investigated, in order to achieve best performance one might have to perform *ad-hoc* adjustments to the solver, depending on the specific sequential operation considered.

Future work will focus on relaxing the limitations outlined above, but also on investigating the applicability of DeepPCR to speedup forward and backward passes through more complex architectures, as well as to speedup different types of sequential operations. In particular, text generation in large language models [4] could be a suitable candidate. Overall, DeepPCR represents a promising method for speeding up training and inference in applications where reducing wall-clock time is critical, and additional computational power is available for parallelization. Furthermore, DeepPCR has the potential to unlock architectures which were not previously experimented upon, due to the long computational time required to perform inference on them.

## Acknowledgements

The authors would like to thank Barry Theobald, David Grangier and Ronan Collobert for their effort and help in proofreading the paper, and Nicholas Apostoloff and Jerremy Holland for supporting this work. The work by Federico Danieli was conducted as part of the AI/ML Residency Program in MLR at Apple.

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

# A Specialization of target system for selected applications

In Sec. 2, we introduced a generic framework for the application of DeepPCR, illustrating how performing a sequential operation can be equivalently recast into solving a block bidiagonal system such as (3). In this section, we further specialize this target system, explicitly recovering its form for the applications considered in our work. Notice that this amounts to prescribing a formula for the step operations $f_l(z)$ in (1), and recovering their Jacobians $J_{f_l}|_z$ appearing in (3).

## A.1 Forward pass in MLPs

The forward pass through a NN can be interpreted as a sequential operation whose $l$-th step $f_l^{FP}(z_{l-1})$ corresponds to the application of layer $l$, to recover activations $z_l$. Considering an MLP for simplicity, the corresponding formula for this operation reads

$$\begin{cases} z_0 = f_0^{FP}(x) = W_0 x + b_0, \\ z_l = f_l^{FP}(z_{l-1}) = W_l \sigma_l(z_{l-1}) + b_l \qquad \text{for} \quad l = 1, \dots, L, \end{cases} \tag{7}$$

for various weights and biases $(W_l, b_l)$, and nonlinear activation functions $\sigma_l(\cdot)$. From (7), we can recover the Jacobian of each step:

$$J_{f_l^{FP}}\Big|_z = W_l \, S_l|_z , \tag{8}$$

where in turn $S_l|_z$ identifies the Jacobian of $\sigma_l(\cdot)$ evaluated at $z$. Since the activation functions are applied element-wise, this reduces to the diagonal matrix

$$S_l|_z = \operatorname*{diag}_i \left\{ \sigma_l'([z]_i) \right\}. \tag{9}$$

Similar considerations can be extended to the forward pass through more complex layers, such as convolutional layers, normalization layers, attention layers, and so on. In those cases, an explicit formula such as (8) for the Jacobian calculations might not be readily available, but one can nonetheless recover it efficiently using `autograd` functionalities.

## A.2 Backward pass in MLPs

The backward pass can also be re-cast as a sequential operation through the layers of a NN, as we illustrate here. The purpose of the backward pass is finding the gradients of a target cost function $c(y, \bar{y})$ with respect to the hidden variables $z_l$, which we can then use to recover the gradients with respect to the network parameters. Considering again a MLP for simplicity, we have that gradients of $c(y, \bar{y})$ with respect to weights and biases at each layer can be recovered as

$$\nabla_{W_l} c = \nabla_{z_l} c \otimes \sigma_l(z_{l-1}), \quad \text{and} \quad \nabla_{b_l} c = \nabla_{z_l} c, \tag{10}$$

with $a \otimes b = a \cdot b^T$ denoting the outer product operator. The gradients $\nabla_{z_l} c$, in turn, are given by applying the chain rule and solving

$$\nabla_{z_{l-1}} c = J_{f_l^{FP}}\Big|_{\hat{z}_{l-1}}^T \nabla_{z_l} c, \quad \text{for} \quad l = 1, \dots, L \tag{11}$$

backwards, starting from $\nabla_{z_L} c$, which can be evaluated directly from the output $y$. This is the target sequential operation that we aim to parallelize. Notice that $\hat{z}_l$ represents the activations at layer $l$ recovered during the forward pass, and that $J_{f_l^{FP}}\Big|_{\hat{z}_{l-1}}^T$ is the transpose of the same operator appearing in (8). Notice moreover that (11) already represents a sequence of *linear* operations, and as such we can solve it by applying PCR directly, without needing Newton iterations. More explicitly, the target system is given by

$$\begin{bmatrix} I & & & \\ -J_{f_L^{FP}}\Big|_{\hat{z}_{L-1}}^T & I & & \\ & \ddots & \ddots & \\ & & -J_{f_1^{FP}}\Big|_{\hat{z}_0}^T & I \end{bmatrix} \begin{bmatrix} \nabla_{z_L} c \\ \nabla_{z_{L-1}} c \\ \vdots \\ \nabla_{z_0} c \end{bmatrix} = \begin{bmatrix} \nabla_{z_L} c \\ 0 \\ \vdots \\ 0 \end{bmatrix}. \tag{12}$$

This property is not limited to MLPs: indeed, backward passes through *any* architecture are linear by default.

### A.3 Forward and backward passes in ResNets

ResNets are characterized by skip connections which relate activations from layers at a specific distance $s$. As such, they break the Markovian assumption that DeepPCR relies on: in fact, the step function representing the application of a layer has form $z_l = f_l^{RN}(z_{l-1}, z_{l-s})$ for $l\%s = 0$ (where $\%$ denotes the modulo operation), meaning it depends both on the activations at the previous layer and those $s$ layers prior. Nonetheless, we can still apply the DeepPCR procedure, if we collapse all layers in-between a skip connection into a single one. This amounts to considering the following step function:

$$
\begin{aligned}
z_l = f_l^{RN}(z_{l-1}, z_{l-s}) &= f_l^{RN}(f_{l-1}^{RN}(z_{l-2}), z_{l-s}) \\
&= \cdots = f_l^{RN}(f_{l-1}^{RN} \circ \cdots \circ f_{l-s+1}^{RN}(z_{l-s}), z_{l-s}) := \hat{f}_l^{RN}(z_{l-s}),
\end{aligned}
\tag{13}
$$

where $\circ$ stands for function composition. Notice that, after this modification is done, one can apply DeepPCR to the reduced system composed only of unknowns $z = \left[z_0^T, z_s^T, z_{2s}^T, \dots\right]$. In our experiments, we consider MLPs as intermediate layers, and simple residual addition as a skip connection layer. This can be denoted as

$$
f_l^{RN} = \begin{cases} W_l \sigma_l(z_{l-1}) + b_l & \text{if} \quad l\%s \neq 0 \\ W_l \sigma_l(z_{l-1}) + b_l + z_{l-s} & \text{else.} \end{cases}
\tag{14}
$$

The Jacobians of the resulting collapsed forward pass is then given by

$$
J_{\hat{f}_l^{RN}}\Big|_{z_l} = I + \prod_{i=0}^{s-1} J_{f_{l-i}^{RN}}\Big|_{z_{l-i}},
\tag{15}
$$

and $J_{f_{l-i}^{RN}}$ shares the same form as $J_{f_{l-i}^{FP}}$ in (8). A similar reasoning can be applied to the backward pass.

### A.4 Denoising procedure in diffusion models

Also image generation in a diffusion model is, at its heart, the result of a sequential operation: an image is extracted from random noise by having it undergo a sequence of denoising steps. As such, this operation too is a target for parallelization by DeepPCR. Following [36], the step function in this case is defined by

$$
z_l = f_l^{DM}(z_{l-1}) = \frac{1}{\sqrt{\alpha_l}}\left(z_l - \frac{1 - \alpha_l}{\sqrt{1 - \bar{\alpha}_l}}g(z_{l-1}, l)\right) + \sqrt{\beta_l}\mathcal{N}(0, 1),
\tag{16}
$$

where $z_l$ denotes a partially denoised image (which underwent $l$ denoising steps), $\mathcal{N}(0, 1)$ is a Normal sample, while $\alpha_l$, $\beta_l$, and $\bar{\alpha}_l$ are parameters of the noise scheduling. Notice that the particularity of this step function is that it hides a NN application inside of it: $g(z, l)$ identifies the action of the NN trained to predict the noise in image $z$ at step $l$. Once again, we only need the Jacobians in order to assemble the target system, which implies that we need to compute the Jacobian of the NN output with respect to its input. We have in fact

$$
J_{f_l^{DM}}\Big|_{z_l} = \frac{I}{\sqrt{\alpha_l}} - \frac{1 - \alpha_l}{\sqrt{1 - \bar{\alpha}_l}} \, J_g|_{(z_{l-1}, l)} \, (z_{l-1}, l).
\tag{17}
$$

Luckily, $J_g|_{(z_{l-1}, l)}$ can be efficiently calculated using `autograd` functionalities, at a cost comparable of that of applying $g(z, l)$.

Notice this applies equivalently to diffusion mechanisms acting in pixel and latent space: in the former, $z_l$ denotes an actual (noisy) image; in the latter, $z_l$ denotes a (noisy) representation of an image, according to a given encoder [37].

## B  Relaxing the Markovian assumption

When introducing our method, in Sec. 2 we remark on the fact that, for DeepPCR to be applicable, we must be tackling a Markovian sequence $z_l = f_l(z_{l-1})$. Indeed, this automatically guarantees the block bidiagonal structure of system (3), upon which the PCR routine relies. In this section,

we comment on the validity of the Markov assumption in applications involving common NN architectures, and provide some workarounds to still guarantee the applicability of DeepPCR in some cases where this assumption does not hold.

As it turns out, many architectures automatically obey the Markovian property: vanilla Feed-forward, Convolutional, and Graph NN all consist in a sequence of layers whose application only depends on the result of the previous layer. This is also the case for uni-directional Recurrent NN; while for bi-directional ones, both the forward and the backward legs, considered separately, are still Markovian. In general, autoregressive models whose state depends only on the previous one (such as the Diffusion Model considered in our experiments), all give rise to Markov sequences which can then directly be tackled by DeepPCR. The main breakdown cases we identified which invalidate the Markovian assumption are twofold: (i) when residual connections are present, and (ii) when considering autoregressive models whose state depends on a *history* of previous states. Even in these cases, though, often one can leverage the available flexibility in defining both the states $z_l$ and the step functions $f_l$ composing the sequence, so to revert back to the Markov case, and still recover a sequence which can be tackled by DeepPCR. In the following, we discuss how this can be achieved for some relevant applications.

**Presence of residual connections**    Residual connections break the Markovian assumption because they introduce dependencies between states far away from each other along the target sequence, so that a certain state $z_l$ ends up becoming a function of multiple previous states $z_l = f_l(z_{l-1}, z_{l-s})$ (see also (14)). Depending on how these connections are distributed, however, different workarounds are available to recover a Markovian sequence:

- If the skip connections length is short with respect to the number of layers ($s \ll L$), one can simply collapse together the states inside each skip, and consider only the sequence composed by the states connected by the skips. This is precisely the approach we utilized in our ResNets applications, so we refer to appendix A.3 for more details on how this can be implemented.

- Conversely, if the skip length is comparable to that of the whole sequence, one can consider applying DeepPCR to parallelize the application of the layers *within* the skip. Notice however that this severely caps the potential for parallelization offered by our method.

- If multiple skip connections are *nested* within each other, that is if we have $z_l = f_l(z_{l-1}, z_{l-s})$ and $z_{\hat{l}} = f_{\hat{l}}(z_{\hat{l}-1}, z_{\hat{l}-\hat{s}})$ for two (or more) given pairs $l - s < \hat{l} - \hat{s} < \hat{l} < l$, then the whole sequence can be cut between $\hat{l} - \hat{s}$ and $\hat{l}$ and split into two parts. Since the residuals originating from the "left" part are not involved in the generation of the left sequence, and similarly for the "right" part, the two parts are Markovian when taken individually. Notice this occurs in UNets, where images at the corresponding resolution level during down- and up-sampling are connected together.

- The failure case for DeepPCR is when each layer is connected to every other, as it happens for DenseNets. In this framework, in fact, the target system (3) would be fully lower-triangular, and it is not possible to leverage any sparsity pattern thereof to return to the bidiagonal (Markovian) case.

**Autoregressive models whose state depends on a *history* of previous states**    If the model relies on a window of $s$ states to produce the next output in the generating sequence (as is the case, for example, with PixelCNN [38]), then we are considering a non-Markovian sequence where $z_l = f_l(z_{l-1}, z_{l-2}, \ldots, z_{l-s})$. To revert to the Markovian case, we can instead consider the collation of states $\hat{z}_{l-1} = \left[ z_{l-1}^T, z_{l-2}^T, \ldots, z_{l-s}^T \right]^T$ and notice that indeed we can write $\hat{z}_l = \hat{f}_l(\hat{z}_{l-1})$. Note that this results in a Jacobian with larger blocks (by a factor $s$), but if the interdependence between the states is limited, it will still present a sparse structure, thus reducing the complexity of the necessary PCR reduction operations.

While the ones outlined in this section are all feasible workarounds to apply DeepPCR when the Markovian assumption is broken, we would like to point out that the solutions proposed should only be treated as indications, since only some of them have been tested in our work. Moreover, as we are shifting away from the Markovian case, the Jacobi approach investigated in [39] should also become competitive: a comparison with DeepPCR within this framework, and a more in-depth investigation of the solutions outlined here is matter of future research.

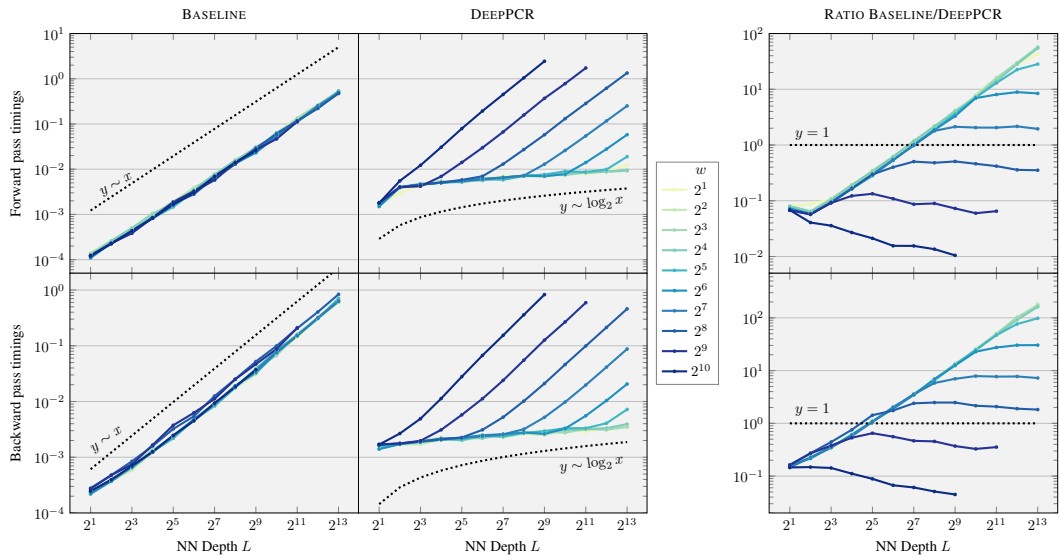

Figure 9: Time to complete a single forward pass (top) and backward pass (bottom), for MLPs of varying depths $L$ and widths $w$. Same configuration as Fig. 3, but using $\tanh$ activation functions. Each datapoint reports the minimum time over 100 runs. The left, center, and right columns refer to the sequential implementation, the DeepPCR implementation, and the ratio between the timings of the two, respectively.

## C  Additional results for single pass in MLPs

In this section, we expand on the results from the application of DeepPCR to accelerate forward and backward passes through MLPs.

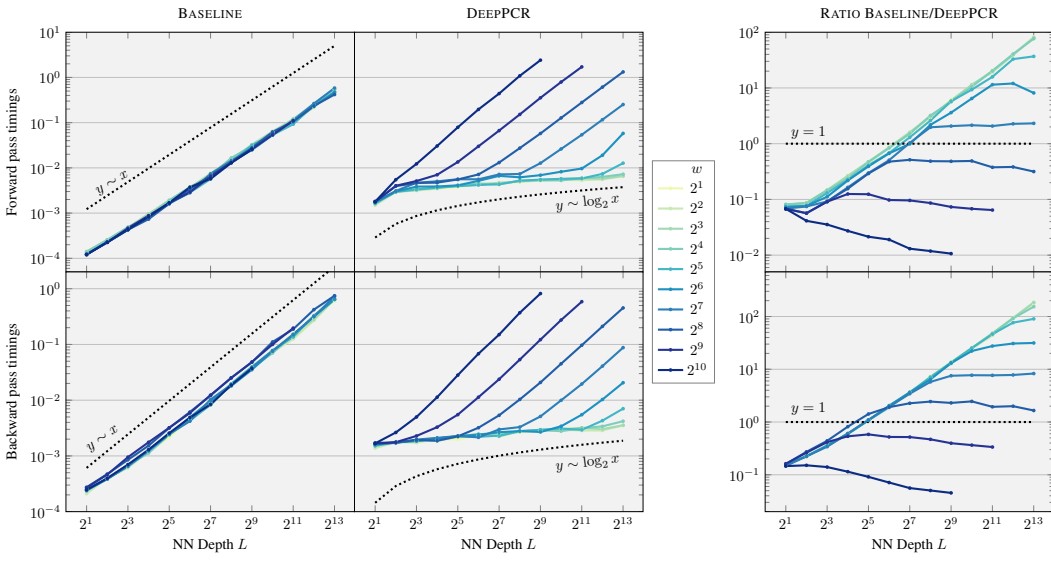

Figure 10: Time to complete a single forward pass (top) and backward pass (bottom), for MLPs of varying depths $L$ and widths $w$. Same configuration as Fig. 3, but using sigmoid activation functions. Each datapoint reports the minimum time over 100 runs. The left, center, and right columns refer to the sequential implementation, the DeepPCR implementation, and the ratio between the timings of the two, respectively.

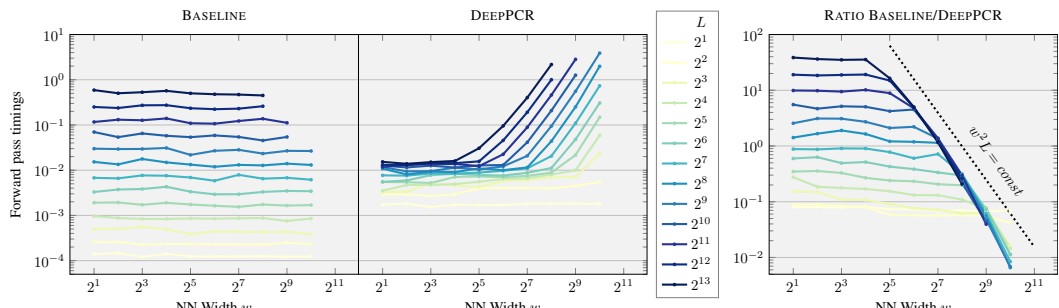

Figure 11: Time to complete a single forward pass, for MLPs of varying depths $L$ and widths $w$. The left column refers to classical sequential implementation, the center column to the parallel implementation using DeepPCR, the right column reports the ratio between the two. Same configuration as Fig. 3, but the results are reported as a function of width, rather than depth.

In Fig. 9 and 10, we report the results from the same experiments in Sec. 4.1, but using $\tanh$ and sigmoid as activation functions, respectively. Notice the overall behaviour is basically indistinguishable from that observed in Fig. 3, proving that DeepPCR manages to speedup the forward pass in MLPs regardless of the nonlinearity considered. Notice moreover that the break-even point for achieving speedup in these cases occurs earlier than with ReLU: at $L$ closer to $2^6$ than $2^7$. This is also in line with what observed in Sec. 4.4, that is that the Newton solver generally requires fewer iterations to reach convergence, when the underlying nonlinearity is smoother.

## C.1 DeepPCR falling to linear regime

In Sec. 4.1, we point out how increasing the values of the parameters $w$ and $L$ past a certain threshold negatively affects the performance of DeepPCR: the middle graph in Fig. 3 shows how, under certain configurations, DeepPCR abandons the logarithmic regime, falling onto a linear one. This is a first indication that the reduction operations in Alg. 1 stop being conducted in parallel, and start being serialized instead. Here we show evidence that this phenomenon is linked to memory limitations in the hardware used for our experiments.

In Fig. 11 we report the same timing results as in Fig. 3, but expressed as a function of the MLPs width $w$ instead. This is to better highlight failure conditions for DeepPCR: since its complexity is only a function of $L$, timings should not be affected by changes in $w$, and hence the curves in Fig. 11 should be constant. This is however not the case, and particularly from the right plot in Fig. 11 we can infer that it is the value of $w^2 L$ that determines when performance drops. To better highlight this, we include in the graph an isoline of $w^2 L$: notice how the threshold after which DeepPCR performance drops has the same inclination. The quantity $w^2 L$ is directly related to the size of the system in (3):

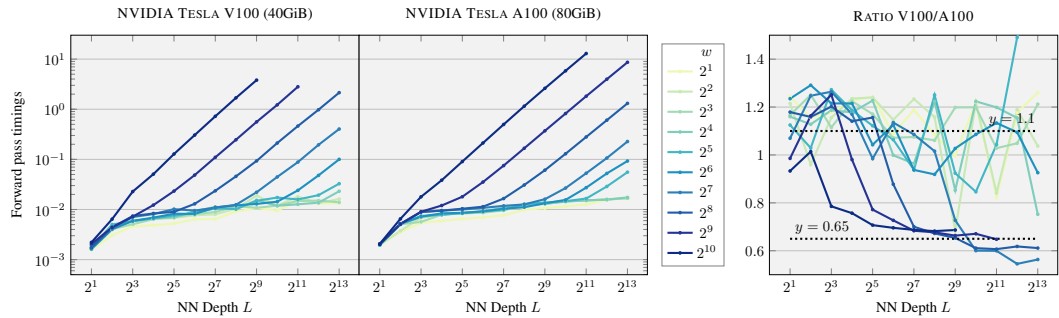

Figure 12: Timings for forward pass through MLP of various sizes $w$, $L$ (same configurations as in Fig. 3), for two different GPUs: V100 (left) and A100 (middle), and ratio between the two (right). The ratios cluster around two values: $1.1$ (showing the GPUs behave comparably for configurations where DeepPCR is still in logarithmic regime), or $0.65$ (when DeepPCR falls to linear regime: the factor of 2 difference stems from the A100 managing to preserve the logarithmic regime for longer).

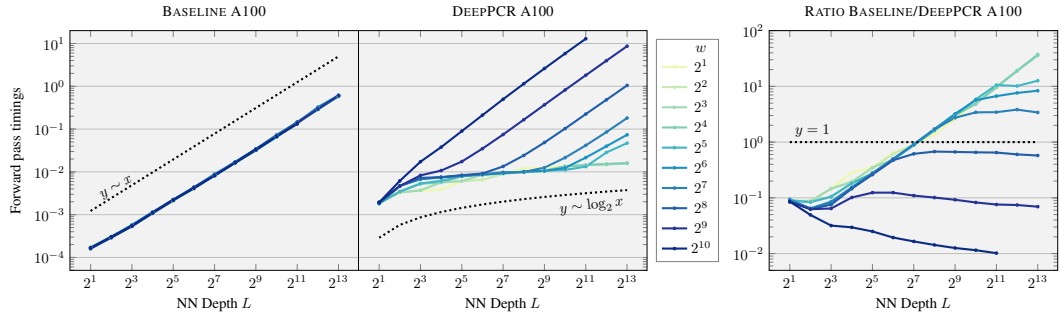

Figure 13: Time to complete a single forward pass, for MLPs of varying depths $L$ and widths $w$, with ReLU activation function. Analogously to Fig. 3, each datapoint reports the minimum time over 100 runs, and the left, center, and right columns refer to the sequential implementation, the DeepPCR implementation, and the ratio between the timings of the two, respectively. The sole difference is that the experiments reported here are conducted on a 80GB A100 GPU.

$w^2$ determines the size of each Jacobian block appearing in the system, while $L$ defines how many blocks there are in total. This hints at the fact that the issue is due to memory limitations. To further verify this, we run the same experiments in Sec. 4.1 on two different Graphical Processing Units (GPUs): an NVIDIA Tesla V100, with 40GB of available memory, and an NVIDIA Tesla A100, with 80GB. We observe that indeed the GPU with larger memory manages to delay the decay of DeepPCR to linear regime by a factor of 2 in the $w^2L$ configuration of the MLP considered. Particularly, this is illustrated in the right plot of Fig. 12, which reports the ratio of the timings for the experiments in Sec. 4.1, conducted with the two GPUs. Notice that the ratios sit around $\sim 1.1$ for the configurations where DeepPCR performance is preserved, and fall to half that value, $\sim 0.65$, when DeepPCR regime falls to linear, further confirming our hypothesis.

For completeness, in Fig. 13 we also report the comparison result of DeepPCR against the sequential baseline (equivalent to the experiment in Fig. 3) on the 80GB A100 GPU: the overall behaviour is comparable to that on the 40GB V100 GPU.

### C.2 Memory consumption in DeepPCR

In Sec. 3.1 we mention the additional memory requirements of DeepPCR, and how these are mostly related to the need of storing intermediate operators $A_l$ resulting from the row reductions in line 6 of Alg. 1. From (4), we can see how these operators have the same size as the Jacobian of the step function $f_l(z)$ of the target sequence considered for parallelization via DeepPCR. When applied

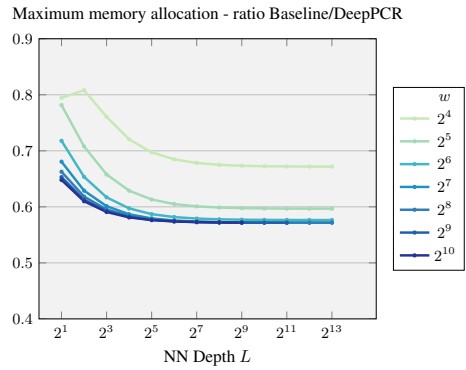

Figure 14: GPU memory allocation for forward pass through MLP: ratio between baseline and DeepPCR for MLPs of varying depths $L$ and widths $w$. Same configurations as in Fig. 3, maximum over 100 runs.

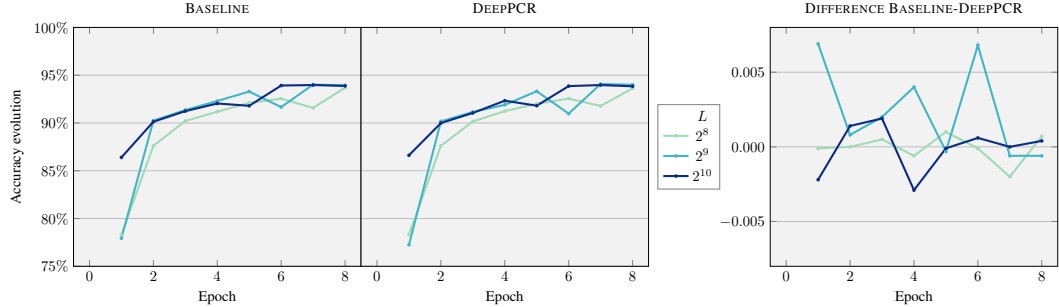

Figure 15: Accuracy evolution during training with forward and backward passes computed sequentially (left) and in parallel with DeepPCR (center), and difference between the two (right), for ResNets of various depths $L$. Same configuration as in Fig. 5.

to parallelize the forward pass through a MLP, as in our experiments in Sec. 4.1, these Jacobians have size $w^2$, exactly as the weight matrices $W_l$ in the network. Since in our experiments storing the model itself is the most memory-consuming operation, we can expect the total memory required by DeepPCR to be roughly twice as much as that required in the sequential approach. This is verified in Fig. 14, where we report the ratio of the maximum memory allocated when performing a forward pass, using the sequential approach and DeepPCR. The ratio sits consistently around $0.6$: in other words, DeepPCR requires slightly less than twice the amount of memory, as expected.

For more general application, one can expect the extra memory requirements of DeepPCR to grow as $w^2L$, $w$ being the size of the Jacobians of the step function $f_l(\boldsymbol{z})$ considered, and $L$ the total number of steps to parallelize over.

## D   Additional results for training of ResNets

In this section, we complement the results in Sec. 4.2, stemming from the applications of DeepPCR to speedup training of ResNets.

In Fig. 15 we report the evolution of the accuracy throughout the training procedure, for the same NN considered in Sec. 4.2. This serves as an additional confirmation that training sequentially or using DeepPCR results in comparable performance for the NN: in both cases, the accuracy hits $94\%$, and their difference is consistently close to $0$.

Additionally, in Fig. 16 we report similar training results stemming from different initializations. Regardless of the seed chosen, the behaviour of both loss and accuracy curves remain consistent, and in particular training sequentially or with DeepPCR still provides comparable results.

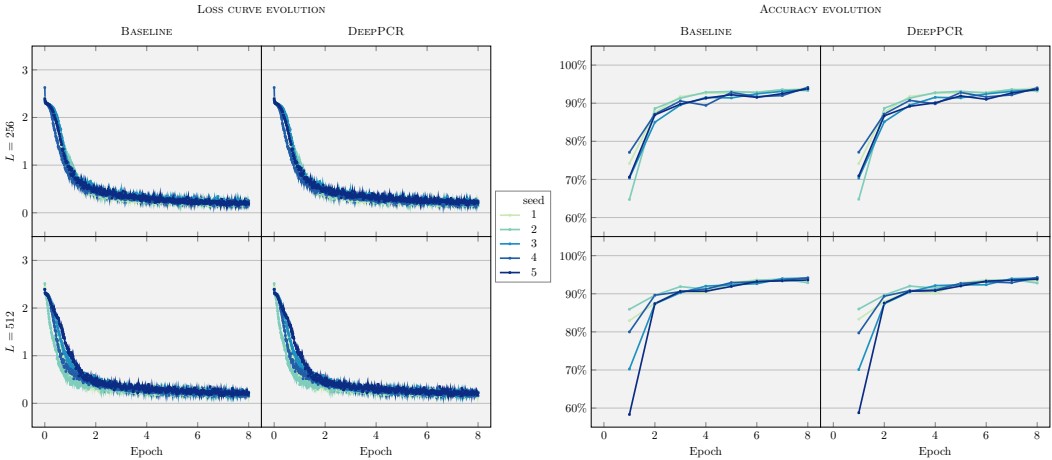

Figure 16: Loss curve and accuracy curve evolution during training for ResNets of different depths, with forward and backward passes computed sequentially and in parallel with DeepPCR. Same configuration as in Fig. 5, but using 5 different seeds for initialization of the network parameters.

# E  Additional results on generation in Diffusion Models

In this section we provide more information regarding the results reported in Sec. 4.3: we describe in detail the architecture of the diffusion models employed, elaborate on how the timings were conducted, and include additional results from pixel-space diffusion, which was not described in the main paper.

## E.1  Choice of architectures and training configuration

To set-up the latent-space diffusion model used for the experiments in Sec. 4.3, we follow Rombach et al. [37] and first train an autoencoder on MNIST by minimizing the loss

$$\mathcal{L}_{AE} = ||x - G(E(x))||_2^2 + \beta \cdot KL(E(x), \mathcal{N}(0, 1)) + \lambda \cdot D(G(E(x))), \tag{18}$$

where $E$ is an encoder, $G$ is a generator/decoder, and $D$ is an adversarial discriminator. We set $\beta$ to $10^-6$ and $\lambda$ to $0.5$ [37]. Differently from Rombach et al. [37], we do not include a perceptual loss for training the autoencoder. Since we are training on MNIST, we also simplify the architectures of $E$, $G$, and $D$, which consist of 4 convolutional layers and produce a 1-dimensional latent space, rather than maintaining the image topology. We experiment with different latent space sizes: $16, 32, 64$, and $128$. Note that similarly to what is done with a variational autoencoder, the latents obtained from the encoder are sampled using the reparameterization trick. Details on the architectures are displayed in Tables 1a and 1b.

Once the autoencoder is trained, we freeze its weights and train a diffusion process on top of the latent vectors produced by applying the encoder on the data. Here, instead of following Rombach et al. [37], we apply the process described in Appendix A.4. Since the latents produced by our autoencoder are 1-dimensional, we use a residual MLP instead of the UNet used by Rombach et al. [37]. See Table 1d for details on the MLP architecture. Finally, we generate new samples by running a diffusion process on randomly sampled latents as in Section 4.3, and generate the final images by applying the decoder just once on the resulting latents. To condition the diffusion model on the current timestep and class, we first embed these two values with a sinusoid positional encoding [42], project them with a 2-layer MLP [37], add them (only if we are conditioning on the class), and pass the result through a linear layer to predict each of the elementwise affine coefficients of the AdaLayerNorm [21] layers of the diffusion MLP.

The models are trained with AdamW with a batch size of $4096$ and a constant learning rate of $10^{-4}$ for $400$ epochs. Following [3], we train the discriminator with the hinge loss and double the learning rate of the generator. We trained all models on a single A100 GPU with 80G of vRAM.

We also experiment with diffusion in pixel space. To this purpose, we train a UNet to recognize the noise in a given image, using MSE as a loss function, following [17]. The training procedure uses AdamW with a constant learning rate of $10^{-4}$ as optimizer, and runs for $500$ epochs with a batch size of $512$. Conditioning with this model is performed in the same way as with the latent-diffusion model, by adding the sinusoidal embedding of time and class. Additional details on the architecture of the network are reported in Tab. 1e.

## E.2  Remarks on profiling for diffusion experiments

**Timing of Jacobian evaluations**  One of the main sub-tasks in the application of DeepPCR for parallelizing the denoising procedure consists in the assembly of the Jacobians of the corresponding step functions (16), and in particular the Jacobians of the NN $g(z, l)$ used as denoiser. This requires assembling one Jacobian per denoising step $l$: a task which is perfectly parallelizable over $l$. To perform this parallelization, we attempted to combine the functionalities provided by `vmap` and `autograd`, which are however a recent addition in PyTorch, and still under development. Indeed, after some testing with this implementation, we noticed that the memory usage during this phase was order of magnitudes higher than actually required, effectively causing the program to crash due to lack of available memory. After some investigation, we suspect the issue is associated with a suboptimal implementation of `vmap` for our target application, which ends up spawning unnecessary copies of the NN, thus rapidly occupying all available memory. To circumvent this issue, and still report a reasonable measure for the performance of DeepPCR, in our experiments we conduct the actual computation of the Jacobians sequentially, and then correct the DeepPCR timing accordingly,

Table 1: Description of latent and pixel diffusion model components. $3 \times 3$ specifies the convolution kernel size. ConvDown is a convolution layer with stride 2, thus reducing spatial resolution by a factor of 2. ConvUp is a convolution layer followed by a nearest-neighbor upsample operation, thus increasing spatial resolution by a factor of 2. AdaLayerNorm corresponds to the 1d version of adaptive instance normalization [21], which re-scales the input based on the diffusion time embedding and an optional class-conditional embedding, following Rombach et al. [37]. We also use Swish activations according to [37]. The input/output dimensionalities `in` and `out` are reported for each layer as `in` $\rightarrow$ `out`. $|Z|$ denotes the latent space dimensionality

(a) **Image Encoder (E)**

Grayscale Image $x \in \mathbb{R}^{32 \times 32}$
$3 \times 3$ ConvDown $1 \rightarrow 16$, ReLU
$3 \times 3$ ConvDown $16 \rightarrow 32$, ReLU
$3 \times 3$ ConvDown $32 \rightarrow 64$, ReLU
$3 \times 3$ ConvDown $64 \rightarrow 128$, ReLU
Reshape $\mathcal{R}^{4 \times 4 \times 128} \rightarrow \mathcal{R}^{2048}$
Linear $2048 \rightarrow 256$
$\mu, \sigma = $ Linear $256 \rightarrow 2 * |Z|$
$z \sim \mathcal{N}(\mu, \sigma)$

(b) **Image decoder/generator (G)**

Latent vector $z \in \mathbb{R}^{|Z|}$
Linear $|Z| \rightarrow 256$
Linear $256 \rightarrow 2048$
Reshape $\mathcal{R}^{2048} \rightarrow \mathcal{R}^{4 \times 4 \times 128}$
$3 \times 3$ ConvUp $128 \rightarrow 64$, ReLU
$3 \times 3$ ConvUp $64 \rightarrow 32$, ReLU
$3 \times 3$ ConvUp $32 \rightarrow 16$, ReLU
$1 \times 1$ Conv $16 \rightarrow 1$, ReLU
Tanh

(c) **Image discriminator (D)**

Grayscale Image $x \in \mathbb{R}^{32 \times 32}$
$3 \times 3$ ConvDown $1 \rightarrow 16$, ReLU
$3 \times 3$ ConvDown $16 \rightarrow 32$, ReLU
$3 \times 3$ ConvDown $32 \rightarrow 64$, ReLU
$3 \times 3$ ConvDown $64 \rightarrow 128$ ReLU
$1 \times 1$ ConvDown $128 \rightarrow 1$

(d) **Residual MLP**

Latent code $z \in \mathbb{R}^{|Z|}$
Linear $|Z| \rightarrow 2048$

| AdaLayerNorm, Swish, Linear $2048 \rightarrow 2048$ AdaLayerNorm, Swish, Linear $2048 \rightarrow 2048$ Residual | $\times 3$ |

LayerNorm, Swish, Linear $2048 \rightarrow |Z|$

(e) **UNet**

Grayscale Image $x \in \mathbb{R}^{w}$
$3 \times 3$ Conv $1 \rightarrow 16$

| $3 \times 3$ Conv $c \rightarrow c$, ReLU, BatchNorm $3 \times 3$ Conv $c \rightarrow c$, ReLU, BatchNorm $3 \times 3$ ConvDown $c \rightarrow 2c$ Residual | $\times 2$ |

| $3 \times 3$ Conv $c \rightarrow c$, ReLU, BatchNorm $3 \times 3$ Conv $c \rightarrow c$, ReLU, BatchNorm $3 \times 3$ ConvUp $c \rightarrow c/2$ Residual | $\times 2$ |

$3 \times 3$ Conv $16 \rightarrow 1$

assuming perfect parallelization. By doing this, we are slightly favoring DeepPCR, as we cannot expect perfect parallelization to hold in practice: there will necessarily be some overhead associated with this operation; however, it is reasonable to assume this overhead to be negligible with respect to the actual assembly of the Jacobians. Moreover, the whole Jacobians assembly operation (which itself can be considered as an overhead associated with the application of DeepPCR) is negligible with respect to the time required to perform the PCR reductions. We report a comparison of these timings in Fig. 17, for the latent diffusion experiments described in Sec. 4.3.

**Ensuring consistency in sequential and DeepPCR denoising procedure** Looking at (16), we can see that denoising procedure is a stochastic process: at each step, random Gaussian noise is added to

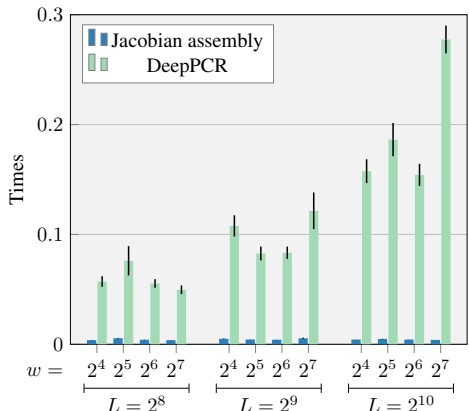

Figure 17: Comparison of the times required for the assembly of the Jacobians (17), and for the complete DeepPCR routine, for the latent diffusion experiments. Setup as in Fig. 6 (left). From this we can infer that overall, the main bottleneck in the application of DeepPCR remains the PCR reduction operations in line 6 of Alg. 1.

the image; this is generally done to ensure variability in the output generated. In our experiments we are interested in matching the results from applying (16) sequentially and those recovered by parallelizing this procedure using DeepPCR. To ensure this, for each generated image we simply pre-sample the noise to include at each step, and feed the same sample to both the sequential and DeepPCR procedures.

### E.3 Additional results on application of DeepPCR to diffusion

In Tab. 2 we expand on the results on latent diffusion in Fig. 6, by reporting numerical values for the timings and speedups recovered, as well as the average number of Newton iterations to convergence and $L^\infty$ norm of the error at convergence. Moreover, we include results from the application of DeepPCR to pixel-space diffusion. Overall, also in this case the convergence error remains small, and the Newton iterations remain bounded, confirming what we observed for Sec. 4.1 and 4.2 as well. Notice however the increase in Newton iterations for the models with the largest sizes, which also has an effect on the speedup recovered: this is investigated more in detail in appendix F.

In Fig. 18 we also show some examples of the images generated by the latent diffusion process trained on MNIST. We put side-by-side the ones recovered using the baseline sequential procedure and those using DeepPCR, to further showcase their similarity.

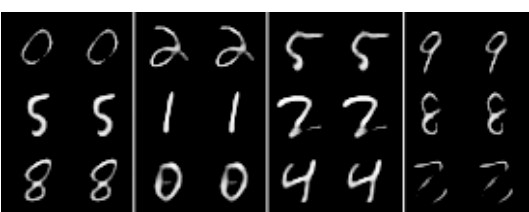

Figure 18: Example images generated by the diffusion models used for the experiments in Fig. 6. The images generated sequentially and using DeepPCR are put side-by-side, and show no visible difference.

Table 2: Times to complete diffusion generation in latent-space and pixel-space (minimum over 100 trials). $\dim(\mathbf{x_0})$ refers to the input dimensions, $L$ is the total number of diffusion steps, SEQUENTIAL is the amount of time inference takes in a sequential fashion, ADJ. DEEPPCR is the amount of time inference takes with DeepPCR: notice this is an estimate, and has been adjusted for parallelism in the Jacobian computation: see details in appendix E.2. We also report the average Newton iterations required (NEWTON ITS.), and the convergence error expressed as the $L^\infty$ norm between the generated image using the sequential model and DeepPCR.

| | $\dim(\mathbf{x_0})$ | $L$ | $\dim(\mathbf{z_1})$ | SEQUENTIAL | ADJ. DEEP-PCR | SPEED-UP | NEWTON ITS. | $L^\infty$ ERROR |
|---|---|---|---|---|---|---|---|---|
| LATENT-SPACE | $32 \times 32$ | 256 | 16 | 0.3063 s | 0.0554 s | 5.53× | 7.05 it | 0.00195 |
| | $32 \times 32$ | 512 | 16 | 0.5997 s | 0.0721 s | 8.32× | 9.06 it | 0.00304 |
| | $32 \times 32$ | 1024 | 16 | 1.2241 s | 0.1091 s | 11.22× | 13.40 it | 0.00523 |
| | $32 \times 32$ | 256 | 32 | 0.3092 s | 0.0547 s | 5.65× | 6.67 it | 0.00252 |
| | $32 \times 32$ | 512 | 32 | 0.6358 s | 0.0757 s | 8.40× | 8.90 it | 0.00298 |
| | $32 \times 32$ | 1024 | 32 | 1.2543 s | 0.1143 s | 10.98× | 14.06 it | 0.00608 |
| | $32 \times 32$ | 256 | 64 | 0.3093 s | 0.0556 s | 5.56× | 6.43 it | 0.00283 |
| | $32 \times 32$ | 512 | 64 | 0.6175 s | 0.0748 s | 8.25× | 9.24 it | 0.00324 |
| | $32 \times 32$ | 1024 | 64 | 1.2509 s | 0.1420 s | 8.81× | 17.35 it | 0.00271 |
| | $32 \times 32$ | 256 | 128 | 0.3151 s | 0.0481 s | 6.55× | 6.09 it | 0.00282 |
| | $32 \times 32$ | 512 | 128 | 0.6303 s | 0.0669 s | 9.42× | 9.08 it | 0.00364 |
| | $32 \times 32$ | 1024 | 128 | 1.2390 s | 0.2033 s | 6.09× | 18.54 it | 0.00248 |
| PIXEL-SPACE | $4 \times 4$ | 256 | 16 | 0.6629 s | 0.0974 s | 6.85× | 6.44 it | 0.00069 |
| | $4 \times 4$ | 512 | 16 | 1.3610 s | 0.1323 s | 10.37× | 8.22 it | 0.00143 |
| | $4 \times 4$ | 1024 | 16 | 2.6695 s | 0.1884 s | 14.25× | 12.33 it | 0.00247 |
| | $8 \times 8$ | 256 | 64 | 0.6971 s | 0.1070 s | 6.55× | 6.89 it | 0.00094 |
| | $8 \times 8$ | 512 | 64 | 1.3112 s | 0.1175 s | 11.18× | 7.89 it | 0.00276 |
| | $8 \times 8$ | 1024 | 64 | 2.8571 s | 0.1792 s | 15.95× | 11.11 it | 0.00418 |
| | $16 \times 16$ | 256 | 256 | 0.6942 s | 0.2018 s | 3.46× | 11.33 it | 0.00126 |
| | $16 \times 16$ | 512 | 256 | 1.3752 s | 0.3940 s | 3.79× | 15.78 it | 0.00235 |
| | $16 \times 16$ | 1024 | 256 | 2.8357 s | 1.1594 s | 2.48× | 27.33 it | 0.02638 |

# F   Dependence of Newton solver on configuration details

As we have seen, a key step of DeepPCR lies in using Newton's method to solve the system of equations (2). In this section we analyze more in detail how changing the Newton solver configuration affects the results in Sec. 4. In particular, we focus on the impact on the solver performance of changing the maximum number of Newton iterations, as well as the choice of initial guess.

## F.1   Changing the maximum number of Newton iterations

Throughout the main text of our paper, we configure the Newton solver with the parameters shown in Tab. 3. From there, we can see that the chosen tolerance is tight, since our aim is to be as accurate as possible, verifying that DeepPCR achieves a solution that differs little from the sequential implementation. One can reasonably expect that decreasing these tolerances will produce a noisier, less-accurate final solution. However, both the stochastic gradient descent algorithm generally used for training NNs, and the denoising procedure used to generate images in diffusion, have a built-in noisy component in their design. We can then consider the approximate results stemming from an early-stopped Newton solver as yet another source of noise.

Table 3: Newton solver parameters and description, along with default values for forward pass computation and diffusion generation.

| PARAMETER | DESCRIPTION | FORWARD PASS | DIFFUSION |
|---|---|---|---|
| NUMBER OF ITERATIONS $c_N$ | Maximum number of Newton iterations the solver will apply. | 15 | 30 |
| ABSOLUTE THRESHOLD | Infinity-norm of the Newton solver residual. | $10^{-4}$ | $10^{-4}$ |
| RELATIVE THRESHOLD | Infinity-norm of the Newton solver residual normalized by the infinity-norm of the first iteration residual. | $10^{-4}$ | $10^{-4}$ |

In the following subsections, we verify the impact of reducing the tolerance in the Newton solver. To this end, we ablate the number of iterations the Newton solver is allowed to take in order to establish whether it is possible to successfully use DeepPCR with a noisier solution for (2): first for ResNets training, and then for diffusion generation.

**Effects on ResNets training**   For these experiments, we set the Newton solver to only stop once the specified number of iterations has been reached $c_N \in \{1, 2, 3, 5, 8, 13\}$, independently of the absolute or relative errors of the residual.

Our results in Fig. 19 show that it is possible to train a ResNet of up to 1024 layers, each layer with a width of 16, and a residual connection every 4 layers, with just 3 Newton iterations for solving each forward pass. This is only half of the 6 iterations we reported in Sec. 4.2 to keep low error with respect to the sequential baseline, and as such represent a potential further $2\times$ speedup with respect to the reported one.

**Effects on Diffusion generation**   In Fig. 20 and 21 we show convergence $L^\infty$ error as a function of the number of Newton iterations, for Diffusion models of 256, 512 and 1024 denoising steps, acting both in latent- and pixel-space. For the former, we consider latent dimension of $w = 16, 32, 64, 128$, while for the latter we consider images of sizes $w = 4 \times 4, 8 \times 8, 16 \times 16$. Each marker in the plots corresponds to a measurement with the maximum number of iterations limited to a certain value.

Notice that, for the most part, the $L^\infty$-norm of the error falls below $0.1$ very quickly (in fewer than 15 iterations regardless of image size). This value already represents a very reasonable convergence, considering that in pixel space the images are scaled within $[-1, 1]$. Moreover, the convergence behaviour is similar regardless of the parameters of the denoising procedure: the convergence curves share similar inclinations.

One exceptions to this is the latent-diffusion model with the most denoising steps in Fig. 20: for the first few Newton iterations there is no improvement of the solution, and convergence starts relatively late in the process. This hints at the fact that a better strategy for initialization might be beneficial in this case (see also appendix F.2). Another exception is in pixel-space diffusion: for the model with largest $L$ in Fig. 20, the Newton solver stalls after reaching $10^{-1}$ error. While this still results in reasonably good solutions, it might represent a bottleneck for scaling the method to pixel-space diffusion in larger dimensions. Further investigation is required in order to identify the causes of this behavior.

## F.2   Changing the initial guess for the Newton solver

Choosing a suitable initial guess $z^0$ for (2) can also have a relevant impact on the performance of the Newton solver. In conducting our experiments, we investigated also this aspect of the solver configuration, and tried to identify ways to initialize the Newton solver which could best leverage the available information. Notice that this additional optimization is only available because DeepPCR resorts to an iterative method to recover the target solution: even if reasonable guesses on the solution are provided, using the sequential approach we cannot, in general, leverage them.

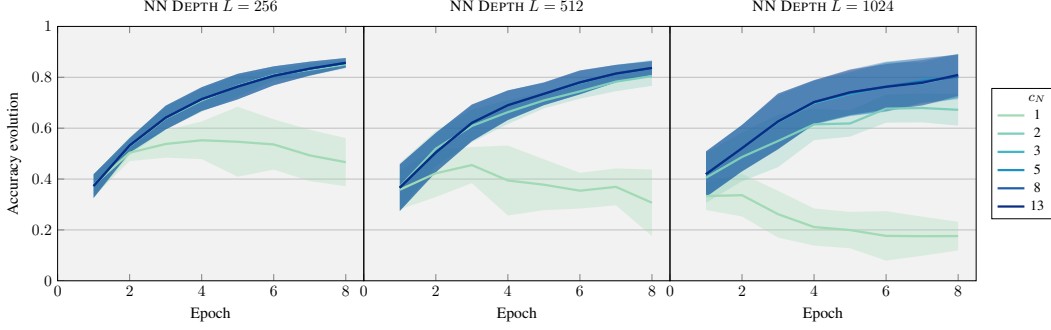

Figure 19: Accuracy of ResNets training using a fixed number of Newton iteration. Lines represent mean accuracy over 5 runs, with shaded area spanning two standard deviations. Notice how, for the curves corresponding to $c_N \geq 3$, the accuracy evolution is basically indistinguishable.

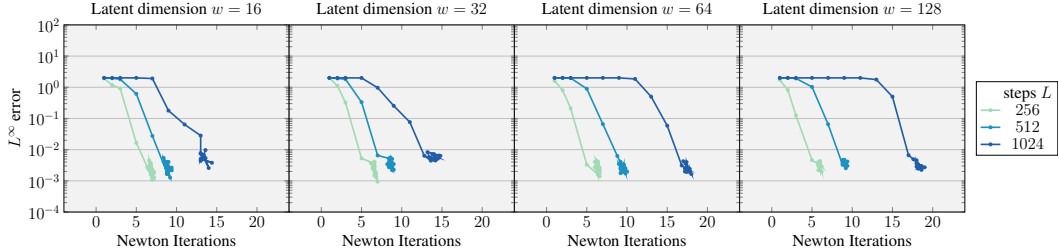

Figure 20: Convergence error in $L^\infty$ norm as a function of the Newton iterations, for latent-space diffusion with models using different latent-space dimensions $w$ and number of denoising steps $L$.

Our choices for the initial guess $z^0$ for the three sets of experiments considered in Sec. 4 are discussed next.

**Forward pass through MLPs**   For the set of experiments in Sec. 4.1, Newton is aimed at recovering the results from the forward pass: namely, the unknowns in $z$ are the activations at each layer. Not much information is available to kickstart the method: as initial guess we simply copy the results from the application of the first layer, but the results (not reported here) are comparable to picking a random initialization. Notice that the backward pass, being a linear operation, does not require Newton iterations, as per the discussion in appendix A.2.

**Training of ResNets**   Also for the set of experiments in Sec. 4.2 we are aiming at recovering the results from the forward pass, but this time this operation is repeated throughout the training procedure: as such, we can reuse information from the previous optimization step to initialize Newton. For the experiments considered, as $z^0$ we use a batch-average of the activations resulting from the forward pass at the previous optimization step. The rationale behind this choice is as follows: at each step of the training procedure, the NN parameters are perturbed only slightly (the SGD update is proportional to the learning rate); moreover, the datapoints in the batches are chosen uniformly, so on average the inputs at each optimization step should be comparable. From these facts, it follows that also the activations should be, on average, comparable.

This choice seems to make for an effective initial guess: in Fig. 22 we measure its impact on the solver convergence, comparing it with zero and random initialization for $z^0$. For the model, we use a similar setup as for Sec. 4.2, with a ResNet network 1024 layers deep, each layer being 16 units wide, with residual connections every 4 layers. The results in Fig. 22 show that we need more than 5 Newton iterations with the zero and random initial guesses, while only 3 suffice when picking as initial guess the average of the activations at the previous optimization step.

**Diffusion generation**   For the set of experiments in Sec. 4.3, the Newton solver is used to recover the noisy images (or their encodings) resulting from each step of the denoising procedure. We can expect the procedure to recover images that are similar to the ones seen during train time, since the

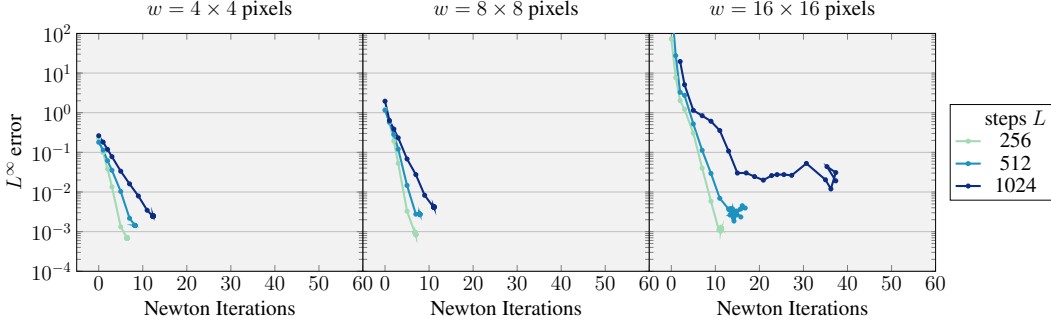

Figure 21: Convergence error in $L^\infty$ norm as a function of the Newton iterations, for pixel-space diffusion with models using different image sizes $w$ and number of denoising steps $L$.

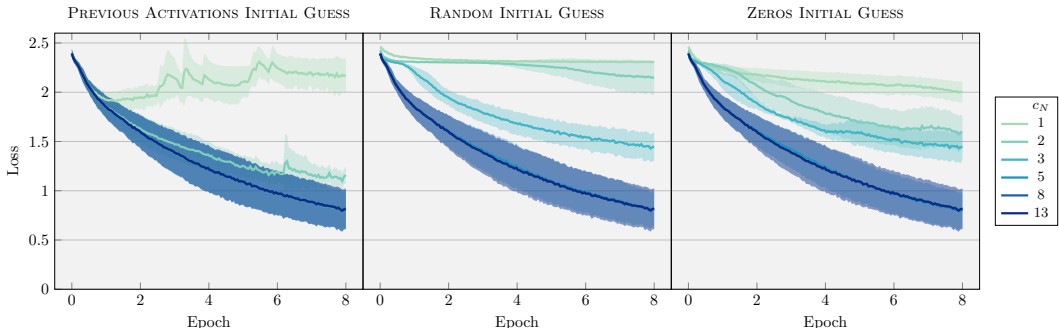

Figure 22: Training dynamics of a 1024 layer ResNet with DeepPCR, different initial guesses, and a fixed set of Newton iterations. Left plot shows shows behaviour when initial guess of the Newton solver is the batch-mean of the activations at the previous optimization step. Middle plot shows the training dynamics when the initial guess is a random guess from a normal distribution. Right plot shows training dynamics when the initial guess is all zeros.

denoiser has been trained for this very purpose. As such, as an initial guess $z^0$ for these noisy images, we pick the average of the train set (either in pixel- or in latent-space).

When considering conditional diffusion, we can further fine-tune the initial guess by taking averages over the images in the train test which belong to the same class. We found that this can generally boost the performance of the Newton solver even further. Results from this comparison are reported in Fig. 23, where we compare the speedup achieved with the conditional and unconditional diffusion models, using their respective initial guesses, for various images sizes. Notice that the conditional models consistently achieve better speedup, thanks to the better initialization for the Newton method.

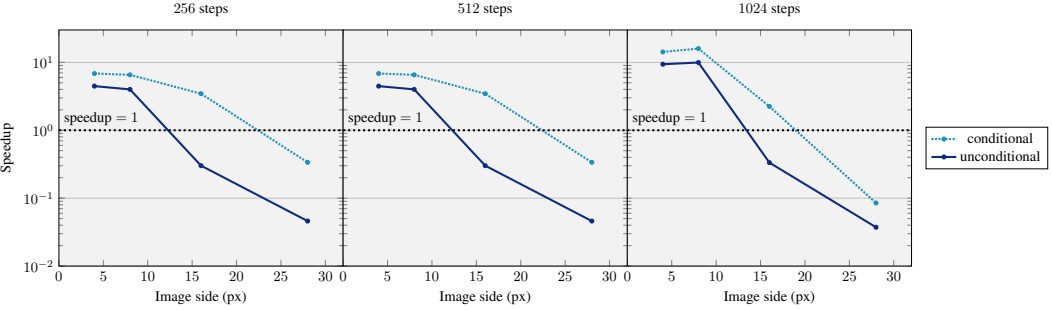

Figure 23: Measured speedup for various pixel diffusion models, as a function of the image size. Notice how conditional models achieve larger speedup due to a more fine-tuned initialization of the Newton method.

