# OpenReview forum: "DeepPCR: Parallelizing Sequential Operations in Neural Networks"
_NeurIPS.cc/2023/Conference — NeurIPS 2023 poster_

### Official Review · Reviewer_7Bpu · 2023-07-06

**Soundness:** 3 good
**Presentation:** 4 excellent
**Contribution:** 3 good
**Rating:** 7
**Confidence:** 3

**Summary:**

This work introduces DeepPCR, a method for parallelizing sequential computations with an application in neural networks. The method is based on the assumption that the target sequence is Markovian, and if so, DeepPCR can leverage Parallel Cyclic Reduction to convert the sequence into a parallel version that executes in O(log_2 L) instead of O(L). To account for the nonlinearity in neural networks, the authors approximate it using Newton’s method. They show that considerable wall-time speedup can be achieved using DeepPCR in both forward and backward passes on MLP/ResNet/diffusion process.

**Strengths:**

* The authors tackle a significant problem using novel approaches. To solve the non-linearity, they propose to use Newton's method and prove that it requires few iterations when the non-linear operations (eg. activation) are smooth.
* The evaluation section was convincing, and the limitations were properly discussed.
* This work could have a profound impact on the parallelization techniques
* The paper was overall well-written, with the main claims clearly articulated and properly supported by references and evaluations.

**Weaknesses:**

* As one of the tradeoffs here is memory v.s. compute time, it would be great to see an analysis of space complexity with respect to the Jacobian blocks/model width, and a detailed memory profile breakdown at training time (which can be obtained using a profiler).
* More evaluations on the transformer-based model will be welcome. The architecture of BERT/T5/GPT should satisfy the Markovian property, and I would expect DeepPCR to be able to work on them.
* To evaluate the quality of stable diffusion models trained using DeepPCR, it would be great to present the FID and the generated images for quantitative and qualitative comparison.

**Questions:**

* One important assumption made by DeepPCR is the Markov property. However, some layers/operators do require additional historical information or outputs of the previous layer. Could you clarify how to handle such cases, for instance, residual connections? Although you may fold the residual blocks(as discussed in the appendix), I wonder what will happen when the sizes of the residual blocks are non-uniform. Will the load of each parallel stage be unbalanced?
* I am interested in the PCR overhead prior to training. How much time is required to perform PCR?


**Limitations:**

The authors have properly addressed the limitations.

---

> ### Author Rebuttal · Authors · 2023-08-09
>
> >W1 As one of the tradeoffs here is memory vs compute time, it would be great to see an analysis of space complexity with respect to the Jacobian blocks/model width, and a detailed memory profile breakdown at training time (which can be obtained using a profiler).
>
> The reviewer raises an excellent point. Due to space limitations, we left the memory complexity analysis in the appendix SecB.2. However, we agree that this should be highlighted in the main text: we hence moved the relevant considerations to Sec3.1. In short, memory usage scales as expected: linearly in the number of steps in the sequence, and quadratically with Jacobian size.
>
> >W2 More evaluations on the transformer-based model will be welcome. The architecture of BERT/T5/GPT should satisfy the Markovian property, and I would expect DeepPCR to be able to work on them
>
> We thank the reviewer for the suggestion: indeed we agree, as this is the focus of our future work. The main complication lies in that generally the text sequence is generated using argmax to sample tokens over a probability distribution. This is heavily nondifferentiable, so Newton is bound to perform poorly. As an alternative, we are considering instead the sequences of probability distributions over the tokens. Nonetheless, identifying the optimal way to apply the algorithm to LLMs remains a matter of future investigation
>
> >W3 To evaluate the quality of stable diffusion models trained using DeepPCR, it would be great to present the FID and the generated images for quantitative and qualitative comparison
>
> Absolutely. Notice that as form of quantitive measure, in Fig6 we considered the _difference_ in Wasserstein distances (same distance used in FID) between baseline and DeepPCR-generated images. This difference is less than 0.01, showing that DeepPCR maintains the same quality of the sequential model. Following the reviewer's feedback, we have included examples of actual images in the appendix (and in the pdf), showing that images generated with DeepPCR are qualitatively indistinguishable from those generated sequentially. We have also included the actual Wasserstein distances of both sequentially and DeepPCR-generated images to the original dataset.
>
> >Q1.1 One important assumption made by DeepPCR is the Markov property. However, some layers/operators do require additional historical information or outputs of the previous layer. Could you clarify how to handle such cases, for instance, residual connections?
>
> We thank the reviewer for the relevant remark. Indeed, the working assumption of our algorithm is that the target sequence $z_l=f_l(z_{l-1})$ is Markovian. Nonetheless, we can leverage the flexibility in defining both the states $z_l$ and the step functions $f_l$ to find workarounds if the Markovian property is not strictly satisfied. These cases are briefly discussed next:
> 1. Residual connections: Depending on how the connections are distributed, feasible workarounds are:
>     * a. Short→ collapse blocks: this is the applied in our ResNets experiments
>     * b. Long→ apply DeepPCR inside, then again hierarchically to stitch the states across the various connections (inspired by [13])
>     * c. Nested (UNets)→ split network in half (downsampling/upsampling), apply DeepPCR to each half. A similar approach can be used for bidirectional RNNs, separating forward and backward part
>     * d. Criss-crossed→collapse blocks and consider state as collation of all states inside
>     * e. DenseNets→failure case of DeepPCR: if all layers are connected to all those after them, the Jacobian is fully lower-triangular, and one cannot leverage any sparsity pattern to return to the bidiagonal/Markovian case.
>
> 2. Autoregressive models relying on a history of previous states: If the model relies on a window of states to produce the next output in the generating sequence (eg PixelCNN), we can collapse these states together to return to the Markovian case, similarly to 1d). Notice this causes each block in the Jacobian to grow, but if the interdependence between the states is limited, it will present a sparse structure, reducing the complexity of its reduction operations
>
> We are grateful to both reviewers Cpwv and 7Bpu for triggering this discussion: we believe it makes for a relevant addition to our work, and we have included an expanded version of it in a novel section of the appendix (to be further refined based on the reviewing process)
>
> > Q1.2 Although you may fold the residual blocks, I wonder what will happen when the sizes of the residual blocks are non-uniform. Will the load of each parallel stage be unbalanced?
>
> Yes, if length/distribution of the residual connections vary throughout the NN, the cost of applying the corresponding step functions $f_l$ also ends up varying with $l$, causing bottlenecks in the application of each step of DeepPCR. However, one can leverage the flexibility in the definition of the sequence $z_l=f_l(z_{l-1})$ to try and render these costs more uniform. For example, by composing together layers including shorter residual connections so to match the cost of applying a layer presenting a longer one
>
> > Q2 I am interested in the PCR overhead prior to training. How much time is required to perform PCR?
>
> Thanks for the relevant remark. The main cost of applying DeepPCR lies in the PCR routine (Alg1). On top of this, the main overhead is associated with assembling system (3), and particularly the Jacobian blocks (but this is a perfectly parallel operation).
>
> The timing results in Sec4 already report the total time required by DeepPCR (overheads included); following the reviewer's suggestion, though, we further break this down to highlight the cost of the Jacobian assembly for the diffusion experiments in Fig6. These remain roughly constant (at ~3.5ms) regardless of the architecture chosen, and low with respect to the total time taken by DeepPCR (which ranges from 0.05s to 0.25s). The figure is included in the pdf, as well as in the main paper in SecD3

---

> > ### Comment · Reviewer_7Bpu · 2023-08-15
> >
> > Thanks for the authors' detailed response!

---

### Official Review · Reviewer_Cpwv · 2023-07-06

**Soundness:** 3 good
**Presentation:** 3 good
**Contribution:** 3 good
**Rating:** 5
**Confidence:** 4

**Summary:**

This paper presents a method called DeepPCR for converting sequential operations in Deep Neural Networks into parallel ones, thus accelerating DNN training, inference, and denoising procedure in diffusion models. The key idea of DeepPCR is to interpret a sequential operation of L steps that satisfies the Markov property as the solution of a system of L equations. DeepPCR tackles this solution by using the Parallel Cyclic Reduction algorithm (PCR) and Newton’s method in parallel, resulting in O(log L) steps compared to O(L) steps of the sequential operations. Evaluation results show DeepPCR brings up to 30X speedup for the forward pass and 200X speedup for the backward pass for certain cases in MLPs, and 11.2X speedup in image generation via diffusion. DeepPCR also accelerates ResNet training by up to 7X.


**Strengths:**

+ This is a solid work that improves the parallel execution of DNNs from a more theoretical and fundamental perspective.
+ This paper is well written. Both the theory and system aspects of DeepPCR are carefully analyzed.
+ The resulting performance gains are significant.


**Weaknesses:**

- The general idea of interpreting a sequential operation as the solution of a large system of equations is not brand new [36].
- It has a relatively strong assumption that the operation sequence needs to satisfy the Markov property, i.e., the output of each step depends only on that of the previous step and no past steps.
- Some evaluation details seem missing or unclear.


**Questions:**

In summary, this is an interesting work that aims to parallelize sequential operations in DNNs from a more theoretical and fundamental perspective. I generally enjoyed my reading and I have some detailed questions as follows besides the items in the weakness section above. It would be very helpful if the authors could provide more justifications:

1. DeepPCR currently works on sequential neural network structures that satisfy the Markov property. Is it a fundamental limitation? It would be helpful to explicitly list some neural architectures that follow (or do not follow) the Markov property to assist in understanding this limitation (if space allows).
2. DeepPCR focuses on execution performance. Some important evaluation specifications are necessary, e.g., GPU type and DNN training/inference framework/engine.
3. DeepPCR trains ResNet on the MNIST dataset. Why not use ImageNet that might be more convincing?
4. What is the relationship between Deep PCR’s design and the effort in operator fusion (and subsequent efficient parallelization)? It would be extremely helpful to include some discussion about this.
5. PCR requires extra memory (and results in more intensive computation) than its sequential counterpart, thus sensitive to the available hardware. How does it affect parallel efficiency? Does this cause any problems for large-scale training tasks that require distributed settings and intensive communications among nodes?


**Limitations:**

Limitations are carefully explained.

---

> ### Author Rebuttal · Authors · 2023-08-09
>
> > W1 The general idea of interpreting a sequential operation as the solution of a large system of equations is not brand new [36]
>
> Indeed this intuition is not new: as mentioned in the paper, the first theorisation of this comes from [30] (a work also referenced in [36]). Still, we do not claim that our novelty lies in this intuition, but rather in the operationalisation of this method for Markovian sequences, as well as an in-depth theoretical and experimental analysis of its characteristics. This application is complementary to the one in [36] (as their method instead targets non-Markovian sequences), but is particularly relevant for modern Deep Learning architectures, as we discuss in W2, Q1
>
> > W2,Q1 DeepPCR currently works on sequential neural network structures that satisfy the Markov property. Is it a fundamental limitation? It would be helpful to explicitly list some neural architectures that follow (or do not follow) the Markov property to assist in understanding this limitation
>
> As correctly stated by the reviewer, our working assumption is that we are dealing with Markovian sequences. We point out that this assumption is typically satisfied: a non-exhaustive list of architectures obeying it is
> * Feed-Forward NN
> * Convolutional NN
> * Graph NN
> * (Unidirectional) Recurrent NN
> * Normalising Flows
> * Diffusion Networks
> * Autoregressive models depending on one previous state
>
> Conversely, this assumption breaks down in two main cases
> * Presence of residual connections
> * Autoregressive models relying on a history of previous states
>
> Even if the Markovian assumption does break down, in some cases one can still rely on some workarounds to apply DeepPCR. Indeed we successfully use one such method in our ResNets experiments. We refer to the answer of Q1 of 7Bpu for a list of possible approaches to circumvent some common non-Markovian cases
>
> > W3, Q2 DeepPCR focuses on execution performance. Some important evaluation specifications are necessary, eg, GPU type and DNN training/inference framework/engine
>
> We thank the reviewer for this remark. We have now included this information in the paper, adding at Line 193: “The experiments in this section were conducted on a V100 GPU with 40GB of RAM; our models are built using the PyTorch framework, without any form of neural network compilation.” For the study on the effect of increasing GPU memory in Fig11, we are using a A100 GPU with 80GB, as reported in Line 540.
> We hope this addresses the reviewer’s concerns. If more details still require clarification, we kindly invite the reviewer to specify them, and we will gladly include them.
>
> > Q3 DeepPCR trains ResNet on the MNIST dataset. Why not use ImageNet that might be more convincing?
>
> We agree with the reviewer on the relevance of showcasing results also on more complex datasets. For the specific experiments in Sec4.1/4.2 our goal is mainly measuring how DeepPCR performance varies in terms of specific architecture parameters (namely, width and depth of the NN considered). A smaller dataset allowed us to iterate freely across these parameters, and since in this case the computations involved are rather agnostic to the type of data being fed to the NN, we do not expect that changing the dataset will affect our conclusions on the recovered speedup.
> Following the reviewer's suggestion, however, we are including additional results on diffusion on CIFAR10 and CelebA. These can be seen in the pdf, and are qualitatively similar to the ones on MNIST, reporting up to 9x and 10x speedups with no loss in FID.
>
> >Q4 What is the relationship between Deep PCR’s design and the effort in operator fusion (and subsequent efficient parallelization)?
>
> We were not familiar with operation fusion, so we are grateful to the reviewer for bringing this to our attention. From [what we could gather](https://learn.microsoft.com/en-us/windows/ai/directml/dml-fused-activations), it is an optimisation to reduce redundant memory reads, merging certain layer operations (most noticeably, the element-wise nonlinearity which typically follows other layers). This could benefit DeepPCR too, as it will accelerate the two main overheads associated with the Newton solver:
> * Residual computation: The right-hand side of the linearised system (3) is computed by applying each “step function” $f_l$ in parallel. These are the same functions constituting the layers of the NN, so these too can benefit from operation fusion, in a similar way as the sequential forward pass
> * Jacobian assembly: Similarly, the Jacobians in (3) are assembled using autograd in parallel over the individual $f_l$: operator fusion should be able to benefit backward passes as well, following similar principles (since also the derivatives of the nonlinearities are an element-wise operation) - if this is true, then DeepPCR can gain performance also in this situation
>
> > Q5 PCR requires extra memory (and results in more intensive computation) than its sequential counterpart, thus sensitive to the available hardware. How does it affect parallel efficiency? Does this cause any problems for large-scale training tasks that require distributed settings and intensive communications among nodes?
>
> We thank the reviewer for raising this interesting point. The inter-node communication overhead associated with the DeepPCR operations ultimately depends on the way the model is partitioned:
> * For a data-parallel setting, no additional communication is necessary, but the GPU must be able to accommodate for the extra memory requirements (roughly $2\times$ the size of the model, as per Line174 and analysis in SecB.2)
> * For a model-parallel setting, there will likely be additional communication during the reduction phase of DeepPCR, depending on how the Jacobians are distributed across the GPUs. Notice that each of the reduction steps (lines 5-6 in Alg1) only involve communication between two nodes ($l$ and $l-i$), although a sync between all nodes is required at the end of each reduction step

---

> > ### Comment · Reviewer_Cpwv · 2023-08-21
> >
> > Thanks for the authors' careful response!

---

> > > ### Author Response · Authors · 2023-08-21
> > >
> > > Thanks again to you for the valuable feedback you’ve provided in your review!
> > >
> > > If you feel like we’ve addressed your concerns in our rebuttal, we would be grateful if you considered increasing the score provided. If not, and some doubts still remain, we would appreciate if you could further point them out, so that we can try and address them properly.

---

### Official Review · Reviewer_uCYF · 2023-07-07

**Soundness:** 3 good
**Presentation:** 3 good
**Contribution:** 2 fair
**Rating:** 5
**Confidence:** 3

**Summary:**

In this paper, the authors proposed an algorithm (DeepPCR) which parallelizes typically sequential operations used in inference and training of neural networks. DeepPCR is based on interpreting a sequence of L steps as the solution of a specific system of equations, which got recovered using the Parallel Cyclic Reduction algorithm.
To verify the the effectiveness of DeepPCR, authors presented the results on multi-layer perceptrons, and reach speedups of up to 30x for forward and 200x for backward pass. In addition, with ResNets with as many as 1024 layers, and generation in diffusion models, DeepPCR enables up to 7x faster training and 11x faster generation, respectively.

**Strengths:**

1. In this paper, authors proposed to interpret a sequential operation of L steps as the solution of a system of L equations, which can be recovered using the Parallel Cyclic Reduction algorithm.
2. The paper is organized and written well.
3. Compared with previous work,  instead of relying on variations of Jacobi iterations,  authors proposed our method specifically targets Markov sequences.
4. In the experiment result section, authors have provided a comprehensive result to demonstrate the applicability of DeepPCR to a variety of scenarios.

**Weaknesses:**

1. The motivation is not very clear. With the constraints of deepPCR, how much real-time saving is not shown in the result. That would be more helpful to understand if the speedup for neural networks sampling on dataset is provided.
2. Compared with Jacobi iterations

**Questions:**

1. Which platform are the experiments in Sec. 4.1 performing at, CPU or GPU? In the appendix, deepPCR rerun for two GPUs: V100 and A100, but didn't see the result of baseline on these two GPUs.
2. Compared with Jacobi iterations in [36], is there any experimental comparison proving deepPCR have more benefit, such as ResNets?

**Limitations:**

As mentioned in the limitation part, the proposed deepPCR has three main limitations:
1. PCR requires fewer sequential steps overall which needs deployments efficiently, otherwise lead to performance degradation.
2. The complexity actually becomes O(cN * log2 L), where cN identifies the number of Newton iterations necessary for convergence. There is a trade-off between complexity and accuracy.
3. Memory increment for the temporary results.

---

> ### Author Rebuttal · Authors · 2023-08-09
>
> > W1.1. The motivation is not very clear.
>
> The main motivation behind the proposed algorithm lies in providing an additional venue for speeding-up training and inference in NNs, to be used in conjunction with already-established parallelisation methods: this is particularly useful in time-sensitive applications, where reducing inference or training time is paramount. Based on the reviewer’s input, we made sure to adapt the abstract to further highlight this, including the sentence “In this work, we introduce DeepPCR, a novel algorithm which parallelizes typically sequential operations in order to speed up inference and training of neural networks", and we remain open to other suggestions.
>
> > W1.2. With the constraints of deepPCR, how much real-time saving is not shown in the result. That would be more helpful to understand if the speedup for neural networks sampling on dataset is provided.
>
> The reviewer raises a valid point. Please notice that in the graphs showcasing our main results (Figs. 3, 4, 6, 8, 10) we do report the wall-times of both baseline and DeepPCR side-by-side. It is true however that in our discussion we mostly focus on speedup: this choice is dictated by the fact that we are mostly interested in checking that the scaling is as predicted by theory. Nonetheless, following the reviewer’s suggestion, we explicitly included some examples or wall-times in the main text as well:
>
> * In Line 205: “Notice this reduces the wall-clock time for a single forward pass from $0.55s$ to $0.015s$, and for a backward pass from $589ms$ to $2.45ms$, corresponding to speedups of $\gt30\times$ and $200\times$, respectively“
> * In Line 235: “Notice that using DeepPCR translates into a speedup of $\sim7\times$ over the sequential implementation: over the whole course of training, this entails a wall-clock time difference of $3.2h$ versus $30min$, even without including the gains from the backward pass.”
> * In Line 264: “We can see that DeepPCR manages to generate images up to $11\times$ faster, reducing the required time from $1.3s$ to $0.12s$”
>
> We hope our reply addresses the reviewer’s concerns. If not, we remain open to provide further clarifications.
>
> > W2, Q2. Compared with Jacobi iterations in [36], is there any experimental comparison proving deepPCR have more benefit, such as ResNets?
>
> We thank the reviewer for pointing this out. We did not consider an experimental comparison between PCR and Jacobi iterations, but this is because the two methods are designed for fundamentally different target applications; as a consequence, we believe a comparison would not be fair (for Jacobi), nor insightful. More specifically, as the authors in [36] point out, Jacobi is performant only if the target system has a “heavy” lower-triangular structure. If instead the system reduces to block bi-diagonal, Jacobi basically becomes equivalent to forward substitution (that is, it reduces to the sequential case), as is described in [36, Sec4.1, Example 3]. Conversely, DeepPCR specifically relies on the block bi-diagonal assumption (guaranteed by the Markovian property of the target sequence), which is why the two algorithms are best suited for fundamentally different scenarios. We discuss this in lines 83-89 of our manuscript.
>
> > Q1. Which platform are the experiments in Sec. 4.1 performing at, CPU or GPU? In the appendix, deepPCR rerun for two GPUs: V100 and A100, but didn't see the result of baseline on these two GPUs.
>
> The experiments in Sec4 were run on a V100 GPU with 40GB of memory. Only the experiments described in Fig11 were run on a A100 with 80GB. The purpose of the latter experiment is to investigate the causes for the lower performance of DeepPCR that we observed in Fig3, and ultimately to pinpoint them to memory limitations - which is why we compare GPUs with different available memory. Since this investigation pertained specifically to the properties of DeepPCR, we initially did not deem relevant to include baseline results as well. However, the reviewer is right in that those results are helpful in providing more evidence of DeepPCR working across different devices. The baseline results have now been included in Fig11, and can be seen in Fig1 in the attached pdf.
> As expected, they are qualitatively similar to the ones recovered on the V100: we believe this further solidifies the theoretical analysis conducted in Sec3.
>
> > L1. PCR requires fewer sequential steps overall which needs deployments efficiently, otherwise lead to performance degradation.
> L2. The complexity actually becomes O(cN * log2 L), where cN identifies the number of Newton iterations necessary for convergence. There is a trade-off between complexity and accuracy.
> L3. Memory increment for the temporary results.
>
> Indeed, these are the main limitations of DeepPCR.
>
> * Regarding L1 and L3: With the experiments in Fig3 and Figs10-11-12, we investigate in detail the effect of memory limitations on DeepPCR, but still identify relevant regimes for speedup.
> * Regarding L2: the trade-off between complexity and accuracy can be seen as both a limitation and an advantage: in our experiments in Sec4.1 we constrain ourselves to maintaining high-accuracy (for a fairer comparison wrt baseline), but one can squeeze additional speedup by sacrificing this, without affecting the final results significantly (this is investigated in Fig.7 and more in detail in App.E)

---

### Author Rebuttal · Authors · 2023-08-09

We are grateful to the reviewers for the comments on our work, and their appreciation for the soundness of the manuscript and for the impactfulness of our proposed algorithm. We are encouraged that all reviewers found that the paper is well-written, that we “provided a comprehensive result to demonstrate the applicability of DeepPCR to a variety of scenarios” (uCYF), that the “resulting performance gains are significant” (CpWv), and that our “work could have a profound impact on the parallelization techniques” (7Bpu). The remarks provided sparkled interesting discussions among the authors, and will undoubtedly contribute to increasing the overall quality of the paper.

Before answering each individual reviewer, we wanted to summarise the main changes that these reviews have triggered:

* We have expanded the results including applications of DeepPCR to Diffusion Models trained on CIFAR10 and on CelebA. The speedups in these cases are consistent with those shown on MNIST, reaching up to 9x and 10x respectively, depending on the architectures considered. These results are provided in the attached pdf and will be included in the updated submission
* We have provided more details about hardware used and the evaluation performed, which have also been included in the paper
* We have listed common architectures that follow the Markovian assumption underlying our algorithm, as well as cases where this assumption does not hold, and identified workarounds that can be used to circumvent this (some of which were tested in our experiments)
* We have better illustrated the relationship between our algorithm and other approaches to speedup such as Jacobi iterations [36] and operators fusion. We have explained in which way they differ and in which case they are complementary to DeepPCR
* Following a reviewer’s suggestion, we have produced a breakdown of the timings required by DeepPCR for the diffusion experiments, showing how overhead costs (particularly, Jacobian assembly) are negligible with respect to the "core" Alg1. These have been included in the pdf, and the updated paper

We appreciate the reviewer’s insights and questions, which we have aimed to address comprehensively. We kindly ask the reviewers for any further feedback they might have, and if they deem appropriate, we would be grateful for taking our responses into account in their assessment.

---

> ### Comment · Area_Chair_cfFv · 2023-08-18
> **Smoothness**
>
> Hi, thank you for your detailed responses.
>
> Could you please comment on the convergence of the Newton’s method? Generally speaking, it is a local method. It would be helpful to discuss its convergence leveraging the special structure of the problem. Under what assumptions does it converge? What tricks (e.g., initialization) are needed to make it converge in practice? I can imagine that if we have a sequence of L hash functions, the method will not work. So some smoothness would be necessary.
>
> It would be nice if the algorithm reduced to the sequential complexity O(L) (and still produced correct results) in these cases instead of not converging / producing wrong results.
>
> Best,
> AC

---

> > ### Author Response · Authors · 2023-08-18
> >
> > > Hi, thank you for your detailed responses.
> > Could you please comment on the convergence of the Newton’s method? Generally speaking, it is a local method. It would be helpful to discuss its convergence leveraging the special structure of the problem. Under what assumptions does it converge? What tricks (e.g., initialization) are needed to make it converge in practice? I can imagine that if we have a sequence of L hash functions, the method will not work. So some smoothness would be necessary.
> >
> > We thank the AC for their question. Indeed, the performance of our method heavily depends on the behaviour of the Newton solver, and investigating its convergence property in this setting is paramount: this is the reason behind our experiments in Sec4.4 and SecE.
> > In general, providing theoretical global convergence guarantees for Newton is not practical (if at all feasible!). Our target system (3) does present a very specific structure, though, which guarantees some useful properties, namely
> > - The Jacobian is always invertible: notice it will always have identities on the main diagonal, hence all its eigenvalues are 1. This implies that the Newton sequence is well-defined
> > - The root is unique and hence has multiplicity one. This stems from the uniqueness of the solution of the forward substitution (sequential) procedure
> >
> > Still, convergence will depend also on the (higher-order derivatives of the) step functions $f_l(z)$ which is ultimately heavily problem- and architecture-dependent. For this reason, we focused more on performing an experimental analysis rather than a theoretical one.
> >
> > In general, for the problems considered we didn’t encounter issues in terms of convergence: this can be seen in Fig7, where Newton iterations remain reasonably bounded even though a non-differentiable function (ReLU) was employed. This extends to the more complex case of diffusion, too (see SecD.3). However, the AC is absolutely correct in that, if we consider extreme cases of non-differentiability (such as argmax or hash functions), the Newton procedure is likely to fail (see also reply to W2 of reviewer 7Bpu). We will make sure to further highlight this in the limitations section.
> >
> > Regarding the role of initialisation, as the AC rightly assumes, we did observe that properly choosing an initial guess for the Newton solver plays a relevant role in reducing the number of iterations to convergence. Nonetheless, for the experiments considered, even employing relatively simple heuristics resulted in reasonably fast convergence: this is investigated in detail in SecE.2. For example, as an initial guess for Newton applied to the training procedure in ResNets we resorted to using the batch-average of the activations at the previous optimisation step, but already with an all-zero initialisation we can observe convergence, albeit slower (see in particular Fig18 for the effect of varying this).
> >
> > > It would be nice if the algorithm reduced to the sequential complexity O(L) (and still produced correct results) in these cases instead of not converging / producing wrong results.
> > Best, AC
> >
> > We thoroughly agree: this would make for an ideal property (and indeed the Jacobi solver in [36] does satisfy this), but it is just not feasible for our choice of solver. This is an expected trade-off for a (generally) faster solver (Jacobi is linear, Newton is quadratic). The classical workaround consists in switching solving strategy depending on the behaviour of the first (few) iterations: if Newton diverges, one can always revert to the sequential solution (although the cost of the first few iterations is indeed wasted in this case). We will explicitly point this out, too, in the paper.
> >
> > We thank again the AC for their questions, and remain at their disposal for any additional clarification.

---

### Decision · Program_Chairs · 2023-09-21

**Decision:**

Accept (poster)

**Comment:**

All reviewers agree that this is an interesting paper.

One of the concerns was that although the current work is inspired by the prior work by Song et al. [36], there is no direct discussion of pros and cons of Newton iteration compared to Jacobi. I think the authors did explain this well in the rebuttal. While there is no global convergence guarantee, it is nice to see that parallel speedup is possible for the strictly sequential case (which is the worst case for Jacobi).